# State-wise Constrained Policy Optimization

**Weiye Zhao**                                                    *weiyezha@andrew.cmu.edu*
*Robotics Institute*
*Carnegie Mellon University*

**Rui Chen**                                                         *ruic3@andrew.cmu.edu*
*Robotics Institute*
*Carnegie Mellon University*

**Yifan Sun**                                                     *yifansu2@andrew.cmu.edu*
*Robotics Institute*
*Carnegie Mellon University*

**Feihan Li**                                                       *feihanl@andrew.cmu.edu*
*Robotics Institute*
*Carnegie Mellon University*

**Tianhao Wei**                                                    *twei2@andrew.cmu.edu*
*Robotics Institute*
*Carnegie Mellon University*

**Changliu Liu**                                                    *cliu6@andrew.cmu.edu*
*Robotics Institute*
*Carnegie Mellon University*

**Reviewed on OpenReview:** *https://openreview.net/forum?id=NgK5etmhz9*

## Abstract

Reinforcement Learning (RL) algorithms have shown tremendous success in simulation environments, but their application to real-world problems faces significant challenges, with safety being a major concern. In particular, enforcing state-wise constraints is essential for many challenging tasks such as autonomous driving and robot manipulation. However, existing safe RL algorithms under the framework of Constrained Markov decision process (CMDP) do not consider state-wise constraints. To address this gap, we propose State-wise Constrained Policy Optimization (SCPO), the first general-purpose policy search algorithm for state-wise constrained reinforcement learning. SCPO provides guarantees for state-wise constraint satisfaction in expectation. In particular, we introduce the framework of Maximum Markov decision process, and prove that the worst-case safety violation is bounded under SCPO. We demonstrate the effectiveness of our approach on training neural network policies for extensive robot locomotion tasks, where the agent must satisfy a variety of state-wise safety constraints. Our results show that SCPO significantly outperforms existing methods and can handle state-wise constraints in high-dimensional robotics tasks.

## 1 Introduction

Reinforcement learning (RL) has achieved remarkable progress in games and control tasks (Mnih et al., 2015; Vinyals et al., 2019; Brown & Sandholm, 2018). However, one major barrier that limits the application of RL algorithms to real-world problems is the lack of safety assurance. RL agents learn to make reward-maximizing decisions, which may violate safety constraints. For example, an RL agent controlling a self-driving car may receive high rewards by driving at high speeds but will be exposed to high chances of collision. Although the

reward signals can be designed to penalize risky behaviors, there is no guarantee for safety. In other words, RL agents may sometimes prioritize maximizing the reward over ensuring safety, which can lead to unsafe or even catastrophic outcomes (Gu et al., 2022).

Emerging in the literature, safe RL aims to provide safety guarantees during or after training. Early attempts have been made under the framework of constrained Markov decision process, where the majority of works enforce cumulative constraints or chance constraints (Ray et al., 2019; Achiam et al., 2017; Liu et al., 2021). In real-world applications, however, many critical constraints are instantaneous. For instance, collision avoidance must be enforced at all times for autonomous cars (Zhao et al., 2023). Another example is that when a robot holds a glass, the robot can only release the glass when the glass is on a stable surface. The violation of those constraints will lead to irreversible failures of the task. In this work, we focus on state-wise (instantaneous) constraints.

The State-wise Constrained Markov decision process (SCMDP) is a novel formulation in reinforcement learning that requires policies to satisfy state-wise constraints. Unlike cumulative or probabilistic constraints, state-wise constraints demand full compliance at each time step as formalized by Zhao et al. (2023). Existing state-wise safe RL methods can be categorized based on whether safety is ensured during training. There is a fundamental limitation that it is impossible to guarantee hard state-wise safety during training without prior knowledge of the dynamic model. In a model-free setting, the more feasible approach is to statistically learn to satisfy state-wise constraints using as few samples as possible. Our paper concentrates on achieving state-wise constraint satisfaction in expectation.

We aim to provide theoretical guarantees on expected state-wise safety violation and worst case reward degradation during training. Our approach is underpinned by a key insight that constraining the maximum violation is equivalent to enforcing state-wise safety. This insight leads to a novel formulation of MDP called the ***Maximum Markov decision process*** (MMDP). With MMDP, we establish a new theoretical result that provides a bound on the difference between the maximum cost of two policies for episodic tasks. This result expands upon the cumulative discounted reward and cost bounds for policy search using trust regions, as previously documented in literature (Achiam et al., 2017). We leverage this result to design a policy improvement step that not only guarantees worst-case performance degradation but also ensures state-wise cost constraints. Our proposed algorithm, ***State-wise Constrained Policy Optimization*** (SCPO), approximates the theoretically-justified update, which achieves a state-of-the-art trade-off between safety and performance. Through experiments, we demonstrate that SCPO effectively trains neural network policies with thousands of parameters on high-dimensional simulated robot locomotion tasks; and is able to optimize rewards while enforcing state-wise safety constraints. This work represents a significant step towards developing practical safe RL algorithms that can be applied to many real-world problems. Our code is available on Github[1].

## 2 Related Work

### 2.1 Cumulative Safety

Cumulative safety requires that the expected discounted return with respect to some cost function is upper-bounded over the entire trajectory. One representative approach is constrained policy optimization (CPO) (Achiam et al., 2017), which builds on a theoretical bound on the difference between the costs of different policies and derives a policy improvement procedure to ensure constraints satisfaction. Another approach is interior-point policy optimization (IPO) (Liu et al., 2019), which augments the reward-maximizing objective with logarithmic barrier functions as penalty functions to accommodate the constraints. Other methods include Lagrangian methods (Ray et al., 2019) which use adaptive penalty coefficients to enforce constraints and projection-based constrained policy optimization (PCPO) (Yang et al., 2020a) which projects trust-region policy updates onto the constraint set. Although our focus is on a different setting of constraints, existing methods are still valuable references for illustrating the advantages of our SCPO. By utilizing MMDP, SCPO breaks the conventional safety-reward trade-off, which results in stronger convergence of state-wise safety constraints and guaranteed performance degradation bounds.

---

[1]https://github.com/intelligent-control-lab/StateWise_Constrained_Policy_Optimization

## 2.2   State-wise Safety

**Hierarchical Policy**   One way to enforce state-wise safety constraints is to use hierarchical policies, with an RL policy generating reward-maximizing actions, and a safety monitor modifying the actions to satisfy state-wise safety constraints (Zhao et al., 2023). Such an approach often requires a perfect safety critic to function well. For example, conservative safety critics (CSC) (Bharadhwaj et al., 2020) propose a safe critic $Q_C(s, a)$, providing a conservative estimate of the likelihood of being unsafe given a state-action pair. If the safety violation exceeds a predefined threshold, a new action is re-sampled from the policy until it passes the safety critic. However, this approach is time-consuming. On the other hand, optimization-based methods such as gradient descent or quadratic programming can be used to find a safe action that satisfies the constraint while staying close to the reference action. Unrolling safety layer (USL) (Zhang et al., 2022b) follows a similar hierarchical structure as CSC but performs gradient descent on the reference action iteratively until the constraint is satisfied based on learned safety critic $Q_C(s, a)$. Finally, instead of using gradient descent, Lyapunov-based policy gradient (LPG) (Chow et al., 2019) and SafeLayer (Dalal et al., 2018) directly solve quadratic programming (QP) to project actions to the safe action set induced by the linearized versions of some learned critic $Q_C(s, a)$. All these approaches suffer from safety violations due to imperfect critic $Q_C(s, a)$, while those solving QPs further suffer from errors due to the linear approximation of the critic. To avoid those issues, we propose SCPO as an end-to-end policy which does not explicitly maintain a safety monitor.

**End-to-End Policy**   End-to-end policies maximize task rewards while ensuring safety at the same time. Related work regarding state-wise safety after convergence has been explored recently. Some approaches (Liang et al., 2018; Tessler et al., 2018) solve a primal-dual optimization problem to satisfy the safety constraint in expectation. However, the associated optimization is hard in practice because the optimization problem changes at every learning step. Bohez et al. (2019) approaches the same setting by augmenting the reward with the sum of the constraint penalty weighted by the Lagrangian multiplier. Although claimed state-wise safety performance, the aforementioned methods do not provide theoretical guarantee and fail to achieve near-zero safety violation in practice. He et al. (2023) proposes AutoCost to automatically find an appropriate cost function using evolutionary search over the space of cost functions as parameterized by a simple neural network. It is empirically shown that the evolved cost functions achieve near-zero safety violation, however, no theoretical guarantee is provided, and extensive computation is required. FAC (Ma et al., 2021) does provide theoretically guaranteed state-wise safety via parameterized Lagrange functions. However, FAC replies on strong assumptions and performs poorly in practice. To resolve the above issues, we propose SCPO as an easy-to-implement and theoretically sound approach with no prior assumptions on the underlying safety functions.

## 3   Problem Formulation

### 3.1   Preliminaries

In this paper, we are especially interested in guaranteeing safety for episodic tasks, which falls within in the scope of finite-horizon Markov decision process (MDP). An MDP is specified by a tuple $(\mathcal{S}, \mathcal{A}, \gamma, R, P, \mu)$, where $\mathcal{S}$ is the state space, and $\mathcal{A}$ is the control space, $R : \mathcal{S} \times \mathcal{A} \mapsto \mathbb{R}$ is the reward function, $0 \leq \gamma < 1$ is the discount factor, $\mu : \mathcal{S} \mapsto \mathbb{R}$ is the initial state distribution, and $P : \mathcal{S} \times \mathcal{A} \times \mathcal{S} \mapsto \mathbb{R}$ is the transition probability function. $P(s'|s, a)$ is the probability of transitioning to state $s'$ given that the previous state was $s$ and the agent took action $a$ at state $s$. A stationary policy $\pi : \mathcal{S} \mapsto \mathcal{P}(\mathcal{A})$ is a map from states to a probability distribution over actions, with $\pi(a|s)$ denoting the probability of selecting action $a$ in state $s$. We denote the set of all stationary policies by $\Pi$. Subsequently, we denote $\pi_\theta$ as the policy that is parameterized by the parameter $\theta$.

The standard goal for MDP is to learn a policy $\pi$ that maximizes a performance measure $\mathcal{J}_0(\pi)$ which is computed via the discounted sum of reward:

$$\mathcal{J}_0(\pi) = \mathbb{E}_{\tau \sim \pi} \left[ \sum_{t=0}^{H} \gamma^t R(s_t, a_t, s_{t+1}) \right], \tag{1}$$

where $H \in \mathbb{N}$ is the horizon, $\tau = [s_0, a_0, s_1, \cdots]$, and $\tau \sim \pi$ is shorthand for that the distribution over trajectories depends on $\pi : s_0 \sim \mu, a_t \sim \pi(\cdot|s_t), s_{t+1} \sim P(\cdot|s_t, a_t)$.

## 3.2 State-wise Constrained Markov Decision Process

A constrained Markov decision process (CMDP) is an MDP augmented with constraints that restrict the set of allowable policies. Specifically, CMDP introduces a set of cost functions, $C_1, C_2, \cdots, C_m$, where $C_i : \mathcal{S} \times \mathcal{A} \times \mathcal{S} \mapsto \mathbb{R}$ maps the state action transition tuple into a cost value. Analogous to equation 1, we denote

$$\mathcal{J}_{C_i}(\pi) = \mathbb{E}_{\tau \sim \pi} \left[ \sum_{t=0}^{H} \gamma^t C_i(s_t, a_t, s_{t+1}) \right], i \in 1, \cdots, m. \tag{2}$$

as the cost measure for policy $\pi$ with respect to cost function $C_i$. Hence, the set of feasible stationary policies for CMDP is then defined as follows, where $d_i \in \mathbb{R}$:

$$\Pi_C = \{ \pi \in \Pi \big| \ \forall i \in 1, \cdots, m, \mathcal{J}_{C_i}(\pi) \le d_i \}. \tag{3}$$

In CMDP, the objective is to select a feasible stationary policy $\pi_\theta$ that maximizes the performance measure:

$$\max_{\pi} \ \mathcal{J}_0(\pi), \ \textbf{s.t.} \ \pi \in \Pi_C. \tag{4}$$

In this paper, we are interested in a special type of CMDP where the safety specification is to persistently satisfy a cost constraint **at every step** (as opposed to constraint of cumulative discounted cost sum over trajectories), which we refer to as *State-wise Constrained Markov decision process* (SCMDP). Like CMDP, SCMDP uses the set of cost functions $C_1, C_2, \cdots, C_m$ to evaluate the instantaneous cost of state action transition tuples. Unlike CMDP, SCMDP requires the cost for every state action transition to satisfy a constraint. Hence, the set of feasible stationary policies for SCMDP is defined as

$$\bar{\Pi}_C = \{ \pi \in \Pi \big| \forall i \in 1, \cdots, m, \ \mathbb{E}_{(s_t, a_t, s_{t+1}) \sim \tau, \tau \sim \pi} \big[ C_i(s_t, a_t, s_{t+1}) \big] \le w_i \} \tag{5}$$

where $w_i \in \mathbb{R}$. Then the objective for SCMDP is to find a feasible stationary policy from $\bar{\Pi}_C$ that maximizes the performance measure. Formally,

$$\max_{\pi} \ \mathcal{J}_0(\pi), \ \textbf{s.t.} \ \pi \in \bar{\Pi}_C \tag{6}$$

The validity of $\bar{\Pi}_C \in \Pi_C$ is demonstrated through the selection of $d_i = w_i \frac{1 - \gamma^{(1+H)}}{1 - \gamma}$ in Equation (4). This indicates that stationary policies feasible for SCMDP are also applicable to CMDP; however, the reverse is not assured. As a result, policies within $\bar{\Pi}_C$ offer a higher level of safety, ensuring that $\mathbb{E}_{(s_t, a_t, s_{t+1}) \sim \tau, \tau \sim \pi} \big[ C_i(s_t, a_t, s_{t+1}) \big]$ for each state is bounded– a condition not guaranteed by CMDP.

## 3.3 Maximum Markov Decision Process

Note that for equation 6, each state-action transition pair introduces a constraint, leading to a complexity that increases nearly cubically as the MDP horizon ($H$) grows, even when tackled by the fastest algorithms (Cohen et al., 2021) (The detailed SCMDP complexity analysis is summarized in Appendix A). Thus it's intractable to solve using conventional reinforcement learning algorithms. Our intuition is that, instead of directly constraining the cost of each possible state-action transition, we can constrain the expected maximum state-wise cost along the trajectory, which is much easier to solve.

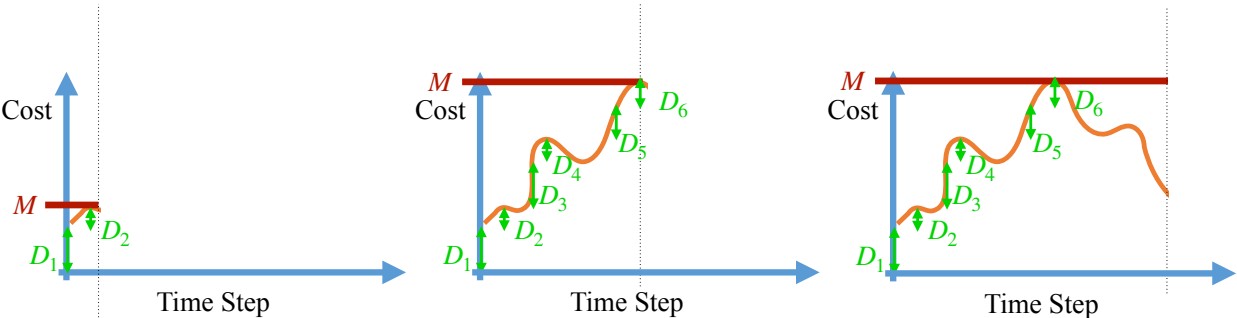

Figure 1: Intuition of the maximum state-wise cost: The three figures above illustrate the evolution of the maximum state-wise cost, denoted as $M$ (shown by the red line), across a single episode. The orange curve represents the state-wise cost, while the green lines with arrows labeled as $D$ indicate the increments of M at each step. Steps with $D = 0$ are not labeled in the figures.

The key challenge lies in efficiently computing the maximum state-wise cost, leveraging the cumulative summation nature inherent in MDP. To achieve this, we introduce a tag $(M)$ that travels along the trajectory, logging the maximum state-wise cost encountered so far. Whenever a higher state-wise cost is identified, $M$ is updated by adding an increment $(D)$, ensuring it consistently reflects the maximum state-wise cost. This tagging mechanism maintains a desirable summation characteristic, facilitating subsequent solutions based on established theoretical results from MDP. This intuition is illustrated in Figure 1

Following that intuition, we define a novel *Maximum Markov decision process* (MMDP), which further extends CMDP via (i) a set of up-to-now maximum state-wise costs $\mathbf{M} \doteq [M_1, M_2, \cdots, M_m]$ where $M_i \in \mathcal{M}_i \subset \mathbb{R}$, and (ii) a set of *cost increment* functions, $D_1, D_2, \cdots, D_m$, where $D_i : (\mathcal{S}, \mathcal{M}_i) \times \mathcal{A} \times \mathcal{S} \mapsto [0, \mathbb{R}^+]$ maps the augmented state action transition tuple into a non-negative cost increment. We define the augmented state $\hat{s} = (s, \mathbf{M}) \in (\mathcal{S}, \mathcal{M}^m) \doteq \hat{\mathcal{S}}$, where $\hat{\mathcal{S}}$ is the augmented state space with $\mathcal{M}^m = (\mathcal{M}_1, \mathcal{M}_2, \cdots, \mathcal{M}_m)$. Formally,

$$D_i(\hat{s}_t, a_t, \hat{s}_{t+1}) = \max\{C_i(s_t, a_t, s_{t+1}) - M_{it}, 0\}, i \in 1, \cdots, m. \tag{7}$$

By setting $D_i(\hat{s}_0, a_0, \hat{s}_1) = C_i(s_0, a_0, s_1)$, we have $M_{it} = \sum_{k=0}^{t-1} D_i(\hat{s}_k, a_k, \hat{s}_{k+1})$ for $t \geq 1$. Hence, we define *expected maximum state-wise cost* (or $D_i$-return) for $\pi$:

$$\mathcal{J}_{D_i}(\pi) = \mathbb{E}_{\tau \sim \pi} \left[ \sum_{t=0}^{H} D_i(\hat{s}_t, a_t, \hat{s}_{t+1}) \right], i \in 1, \cdots, m. \tag{8}$$

Importantly, equation 8 is the key component of MMDP and differs our work from existing safe RL approaches that are based on CMDP cost measure equation 2. With equation 8, equation 6 can be rewritten as:

$$\max_{\pi} \mathcal{J}(\pi), \ \textbf{s.t.} \ \forall i \in 1, \cdots, m, \mathcal{J}_{D_i}(\pi) \leq w_i, \tag{9}$$

where $\mathcal{J}(\pi) = \mathbb{E}_{\tau \sim \pi} \left[ \sum_{t=0}^{H} \gamma^t R(\hat{s}_t, a_t, \hat{s}_{t+1}) \right]$ and $R(\hat{s}, a, \hat{s}') \doteq R(s, a, s')$. With $R(\tau)$ being the discounted return of a trajectory, we define the on-policy value function as $V^\pi(\hat{s}) \doteq \mathbb{E}_{\tau \sim \pi}[R(\tau)|\hat{s}_0 = \hat{s}]$, the on-policy action-value function as $Q^\pi(\hat{s}, a) \doteq \mathbb{E}_{\tau \sim \pi}[R(\tau)|\hat{s}_0 = \hat{s}, a_0 = a]$, and the advantage function as $A^\pi(\hat{s}, a) \doteq Q^\pi(\hat{s}, a) - V^\pi(\hat{s})$. Lastly, we define on-policy value functions, action-value functions, and advantage functions for the cost increments in analogy to $V^\pi$, $Q^\pi$, and $A^\pi$, with $D_i$ replacing $R$, respectively. We denote those by $V_{D_i}^\pi$, $Q_{D_i}^\pi$ and $A_{D_i}^\pi$.

## 4 State-wise Constrained Policy Optimization

To solve large and continuous MDPs, policy search algorithms search for the optimal policy within a set $\Pi_\theta \subset \Pi$ of parametrized policies. In local policy search (Peters & Schaal, 2008), the policy is iteratively updated by maximizing $\mathcal{J}(\pi)$ over a local neighborhood of the most recent policy $\pi_k$. In local policy search

for SCMDPs, policy iterates must be feasible, so optimization is over $\Pi_\theta \bigcap \bar{\Pi}_C$. The optimization problem is:

$$\pi_{k+1} = \underset{\pi \in \Pi_\theta}{\mathbf{argmax}} \; \mathcal{J}(\pi), \tag{10}$$

$$\mathbf{s.t.} \; \mathcal{D}ist(\pi, \pi_k) \leq \delta,$$

$$\mathcal{J}_{D_i}(\pi) \leq w_i, i = 1, \cdots, m.$$

where $\mathcal{D}ist$ is some distance measure, and $\delta > 0$ is a step size. For actual implementation, we need to evaluate the constraints first in order to determine the feasible set. However, it is challenging to evaluate the constraints using samples during the learning process. In this work, we propose SCPO inspired by trust region optimization methods (Schulman et al., 2015). SCPO approximates equation 10 using (i) KL divergence distance metric $\mathcal{D}ist$ and (ii) surrogate functions for the objective and constraints, which can be easily estimated from samples on $\pi_k$. Mathematically, SCPO requires the policy update at each iteration to be bounded within a trust region, and updates policy via solving the following optimization:

$$\pi_{k+1} = \underset{\pi \in \Pi_\theta}{\mathbf{argmax}} \; \underset{\substack{\hat{s} \sim d^{\pi_k} \\ a \sim \pi}}{\mathbb{E}} [A^{\pi_k}(\hat{s}, a)] \tag{11}$$

$$\mathbf{s.t.} \quad \mathbb{E}_{\hat{s} \sim \bar{d}^{\pi_k}}[\mathcal{D}_{KL}(\pi \| \pi_k)[\hat{s}]] \leq \delta,$$

$$\mathcal{J}_{D_i}(\pi_k) + \underset{\substack{\hat{s} \sim \bar{d}^{\pi_k} \\ a \sim \pi}}{\mathbb{E}} \left[ A^{\pi_k}_{D_i}(\hat{s}, a) \right] + 2(H+1)\epsilon^\pi_{D_i} \sqrt{\frac{1}{2}\delta} \leq w_i, i = 1, \cdots, m.$$

where $\mathcal{D}_{KL}(\pi' \| \pi)[\hat{s}]$ is KL divergence between two policy $(\pi', \pi)$ at state $\hat{s}$, the set $\{\pi \in \Pi_\theta : \mathbb{E}_{\hat{s} \sim \bar{d}^{\pi_k}}[\mathcal{D}_{KL}(\pi \| \pi_k)[\hat{s}]] \leq \delta\}$ is called *trust region*, $d^{\pi_k} \doteq (1-\gamma)\sum_{t=0}^{H} \gamma^t P(\hat{s}_t = \hat{s}|\pi_k)$, $\bar{d}^{\pi_k} \doteq \sum_{t=0}^{H} P(\hat{s}_t = \hat{s}|\pi_k)$ and $\epsilon^\pi_{D_i} \doteq \mathbf{max}_{\hat{s}}|\mathbb{E}_{a \sim \pi}[A^{\pi_k}_{D_i}(\hat{s}, a)]|$. We then show that SCPO guarantees (i) worst case maximum state-wise cost violation, and (ii) worst case performance degradation for policy update, by establishing new bounds on the difference in returns between two stochastic policies $\pi$ and $\pi'$ for MMDPs.

**Theoretical Guarantees for SCPO**    We start with the theoretical foundation for our approach, i.e. a new bound on the difference in state-wise maximum cost between two arbitrary policies. The following theorem connects the difference in maximum state-wise cost between two arbitrary policies to the total variation divergence between them. Here total variation divergence between discrete probability distributions $p, q$ is defined as $\mathcal{D}_{TV}(p\|q) = \frac{1}{2}\sum_i |p_i - q_i|$. This measure can be easily extended to continuous states and actions by replacing the sums with integrals. Thus, the total variation divergence between two policy $(\pi', \pi)$ at state $\hat{s}$ is defined as: $\mathcal{D}_{TV}(\pi' \| \pi)[\hat{s}] = \mathcal{D}_{TV}(\pi'(\cdot|\hat{s}) \| \pi(\cdot|\hat{s}))$.

**Theorem 1** (Trust Region Update State-wise Maximum Cost Bound). *For any policies $\pi', \pi$, with $\epsilon^{\pi'}_D \doteq \mathbf{max}_{\hat{s}}|\mathbb{E}_{a \sim \pi'}[A^\pi_D(\hat{s}, a)]|$, and define $\bar{d}^\pi = \sum_{t=0}^{H} P(\hat{s}_t = \hat{s}|\pi)$ as the non-discounted augmented state distribution using $\pi$, then the following bound holds:*

$$\mathcal{J}_D(\pi') - \mathcal{J}_D(\pi) \leq \underset{\substack{\hat{s} \sim \bar{d}^\pi \\ a \sim \pi'}}{\mathbb{E}} \left[ A^\pi_D(\hat{s}, a) + 2(H+1)\epsilon^{\pi'}_D \mathcal{D}_{TV}(\pi' \| \pi)[\hat{s}] \right]. \tag{12}$$

The proof for Theorem 1 is summarized in Appendix C. Next, we note the following relationship between the total variation divergence and the KL divergence (Boyd et al., 2003; Achiam et al., 2017): $\mathbb{E}_{\hat{s} \sim \bar{d}^\pi}[\mathcal{D}_{TV}(p\|q)[\hat{s}]] \leq \sqrt{\frac{1}{2}\mathbb{E}_{\hat{s} \sim \bar{d}^\pi}[\mathcal{D}_{KL}(p\|q)[\hat{s}]]}$. The following bound then follows directly from Theorem 1:

$$\mathcal{J}_D(\pi') \leq \mathcal{J}_D(\pi) + \underset{\substack{\hat{s} \sim \bar{d}^\pi \\ a \sim \pi'}}{\mathbb{E}} \left[ A^\pi_D(\hat{s}, a) + 2(H+1)\epsilon^{\pi'}_D \sqrt{\frac{1}{2}\mathbb{E}_{\hat{s} \sim \bar{d}^\pi}[\mathcal{D}_{KL}(\pi' \| \pi)[\hat{s}]]} \right]. \tag{13}$$

By Equation (13), we have a guarantee for satisfaction of maximum state-wise constraints:

**Proposition 1** (SCPO Update Constraint Satisfaction). *Suppose $\pi_k, \pi_{k+1}$ are related by equation 11, then $D_i$-return for $\pi_{k+1}$ satisfies*

$$\forall i \in 1, \cdots, m, \mathcal{J}_{D_i}(\pi_{k+1}) \leq w_i.$$

Proposition 1 is the first finite-horizon variant of the policy improvement theorem, and it also presents the first constraint satisfaction guarantee under MMDP. Unlike trust region methods such as CPO and TRPO, which assume a discounted infinite horizon sum characteristic, MMDP's non-discounted finite horizon sum characteristic invalidates these theories and separate treatment is required. As the maximum state-wise cost is calculated through a summation of non-discounted increments, analysis must be performed on a finite horizon to upper bound the worst-case summation.

Next, we provide the performance guarantee of SCPO. Previous analyses of performance guarantees have focused on infinite-horizon MDP. We generalize the analysis to finite-horizon MDP, inspired by previous work (Kakade & Langford, 2002; Schulman et al., 2015; Achiam et al., 2017), and prove it in Appendix D. The infinite-horizon case can be viewed as a special case of the finite-horizon setting.

**Proposition 2** (SCPO Update Worst Performance Degradation). *Suppose $\pi_k, \pi_{k+1}$ are related by equation 11, with $\epsilon^{\pi_{k+1}} \doteq \max_{\hat{s}} |\mathbb{E}_{a \sim \pi_{k+1}}[A^{\pi_k}(\hat{s}, a)]|$, then performance return for $\pi_{k+1}$ satisfies*

$$\mathcal{J}(\pi_{k+1}) - \mathcal{J}(\pi_k) > -\frac{\sqrt{2\delta}\gamma\epsilon^{\pi_{k+1}}}{1 - \gamma}.$$

Proposition 2 establishes a fundamental result that bounds the performance degradation when policy updates are carried out via solving equation 10, which ensures satisfaction of the trust region step size constraint and the state-wise maximum cost constraints. Intuitively, this proposition assures that when our policy is updated within these specified constraints, the degradation in reward performance will be limited. This means that our approach strikes a balance between improving the policy's performance and satisfying the state-wise safety constraints.

## 5 Practical Implementation

In this section, we show how to (a) implement an efficient approximation to the update equation 11, (b) encourage learning even when equation 11 becomes infeasible, and (c) handle the difficulty of fitting augmented value $V_{D_i}^{\pi}$ which is unique to our novel MMDP formulation. The full SCPO pseudocode is given as algorithm 1 in appendix E.

**Practical implementation with sample-based estimation** We first estimate the objective and constraints in equation 11 using samples. Note that we can replace the expected advantage on rewards using an importance sampling estimator with a sampling distribution $\pi_k$ (Achiam et al., 2017) as

$$\mathbb{E}_{\hat{s} \sim d^{\pi_k}, \ a \sim \pi}[A^{\pi_k}(\hat{s}, a)] = \mathbb{E}_{\hat{s} \sim d^{\pi_k}, \ a \sim \pi_k}\left[\frac{\pi(a|\hat{s})}{\pi_k(a|\hat{s})}A^{\pi_k}(\hat{s}, a)\right]. \tag{14}$$

equation 14 allows us to replace $A^{\pi_k}$ with empirical estimates at each state-action pair $(\hat{s}, a)$ from rollouts by the previous policy $\pi_k$. The empirical estimate of reward advantage is given by $R(\hat{s}, a, \hat{s}') + \gamma V^{\pi_k}(\hat{s}') - V^{\pi_k}(\hat{s})$. $V^{\pi_k}(\hat{s})$ can be computed at each augmented state by taking the discounted future return. The same can be applied to the expected advantage with respect to cost increments, with the sample estimates given by $D_i(\hat{s}, a, \hat{s}') + V_{D_i}^{\pi_k}(\hat{s}') - V_{D_i}^{\pi_k}(\hat{s})$. $V_{D_i}^{\pi_k}(\hat{s})$ is computed by taking the non-discounted future $D_i$-return. To proceed, we convexify equation 11 by approximating the objective and cost constraint via first-order expansions, and the trust region constraint via second-order expansions. Then, equation 11 can be efficiently solved using duality (Achiam et al., 2017).

**Infeasible constraints** An update to $\theta$ is computed every time equation 11 is solved. However, due to approximation errors, sometimes equation 11 can become infeasible. In that case, we follow Achiam et al. (2017) to propose an recovery update that only decreases the constraint value within the trust region. In addition, approximation errors can also cause the proposed policy update (either feasible or recovery) to violate the original constraints in equation 11. Hence, each policy update is followed by a backtracking line search to ensure constraint satisfaction. If all these fails, we relax the search condition by also accepting decreasing expected advantage with respect to the costs, when the cost constraints are already violated. Denoting $c_i \doteq \mathcal{J}_{D_i}(\pi_k) + 2(H+1)\epsilon_D^\pi \sqrt{\delta/2} - w_i$, the above criteria can be summarized as

$$\mathbb{E}_{\hat{s}\sim\bar{d}^{\pi_k}}[\mathcal{D}_{KL}(\pi\|\pi_k)[\hat{s}]] \leq \delta \tag{15}$$

$$\mathbb{E}_{\hat{s}\sim\bar{d}^{\pi_k},a\sim\pi}\left[A_{D_i}^{\pi_k}(\hat{s},a)\right] - \mathbb{E}_{\hat{s}\sim\bar{d}^{\pi_k},a\sim\pi_k}\left[A_{D_i}^{\pi_k}(\hat{s},a)\right] \leq \max(-c_i, 0), i \in 1, \cdots, m. \tag{16}$$

Note that the previous expected advantage $\mathbb{E}_{\hat{s}\sim\bar{d}^{\pi_k},a\sim\pi_k}\left[A_{D_i}^{\pi_k}(\hat{s},a)\right]$ is also estimated from rollouts by $\pi_k$ and converges to zero asymptotically, which recovers the original cost constraints in equation 11.

**Imbalanced cost value targets** A critical step in solving equation 11 is to fit the cost increment value functions $V_{D_i}^{\pi_k}(\hat{s}_t)$. By definition, $V_{D_i}^{\pi_k}(\hat{s}_t)$ is equal to the maximum cost increment in any future state over the maximal state-wise cost so far. In other words, the true $V_{D_i}^{\pi_k}$ will always be zero for all $\hat{s}_{t:H}$ when the maximal state-wise cost has already occurred before time $t$. In practice, this causes the distribution of cost increment value function to be highly zero-skewed and makes the fitting very hard. To mitigate the problem, we sub-sample the zero-valued targets to match the population of non-zero values. We provide more analysis on this trick in Q3 in section 6.2.

## 6 Experiments

In our experiments, we aim to answer these questions:

**Q1** How does SCPO compare with other state-of-the-art methods for safe RL?

**Q2** What benefits are demonstrated by constraining the maximum state-wise cost?

**Q3** How does the sub-sampling trick of SCPO impact its performance? Does sub-sampling work for other baselines?

**Q4** How tight is our derived surrogate function?

**Q5** How does the resource usage of our algorithm compare to other algorithms?

### 6.1 Experiment Setups

**New Safety Gym** To showcase the effectiveness of our state-wise constrained policy optimization approach, we enhance the widely recognized safe reinforcement learning benchmark environment, Safety Gym (Ray et al., 2019), by incorporating additional robots and constraints. Subsequently, we perform a series of experiments on this augmented environment.

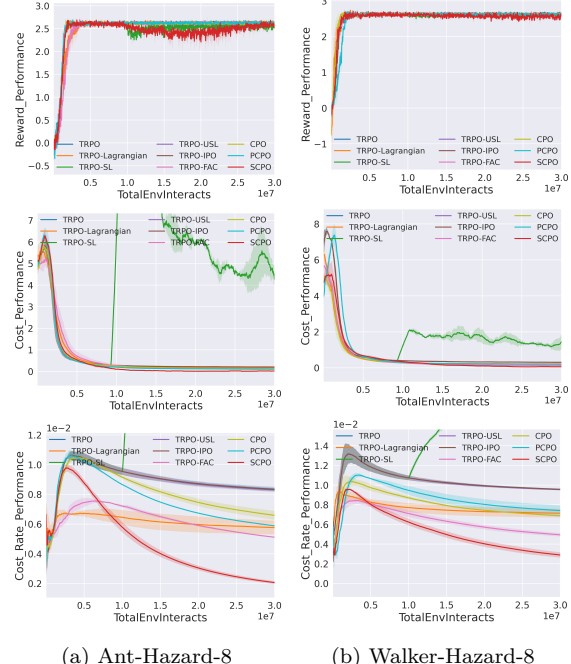

(a) Ant-Hazard-8  (b) Walker-Hazard-8

Figure 2: Comparison of results from two representative test suites in high dimensional systems (Ant and Walker).

Our experiments are based on five different robots: (i) **Point:** Figure 3a A point-mass robot ($\mathcal{A} \subseteq \mathbb{R}^2$) that can move on the ground. (ii) **Swimmer:** Figure 3b A three-link robot ($\mathcal{A} \subseteq \mathbb{R}^2$) that can move on the ground. (iii) **Walker:** Figure 3d A bipedal robot ($\mathcal{A} \subseteq \mathbb{R}^{10}$) that can move on the ground. (iv) **Ant:** Figure 3c A quadrupedal robot ($\mathcal{A} \subseteq \mathbb{R}^8$) that can move on the ground. (v) **Drone:** Figure 3e A quadrotor robot

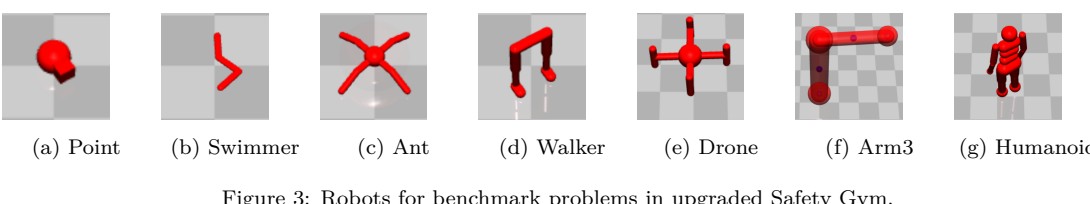

| (a) Point | (b) Swimmer | (c) Ant | (d) Walker | (e) Drone | (f) Arm3 | (g) Humanoid |

Figure 3: Robots for benchmark problems in upgraded Safety Gym.

| (a) Hazard | (b) 3D Hazard | (c) Pillar |

Figure 4: Constraints for benchmark problems in upgraded Safety Gym.

($\mathcal{A} \subseteq \mathbb{R}^4$) that can move in the air. (vi) **Arm3:** Figure 3f A fixed three-joint robot arm($\mathcal{A} \subseteq \mathbb{R}^3$) that can move its end effector around with high flexibility. (vii) **Humanoid:** Figure 3g A bipedal robot($\mathcal{A} \subseteq \mathbb{R}^{17}$) that has a torso with a pair of legs and arms.

All of the experiments are based on the goal task where the robot must navigate to a goal. Additionally, since we are interested in episodic tasks (finite-horizon MDP), the environment will be reset once the goal is reached. For the robots that can move in 3D spaces (e.g, the Drone robot, Arm3 robot), we also design a new 3D goal task with a sphere goal floating in the 3D space. Three different types of constraints are considered: (i) **Hazard**: Dangerous areas as shown in Figure 4a. Hazards are trespassable circles on the ground. The agent is penalized for entering them. (ii) **3D Hazard**: 3D Dangerous areas as shown in Figure 4b. 3D Hazards are trespassable spheres in the air. The agent is penalized for entering them. (iii) **Pillar**: Fixed obstacles as shown in Figure 4c. The agent is penalized for hitting them.

Considering different robots, constraint types, and constraint difficulty levels, we design 14 test suites with 5 types of robots and 9 types of constraints, which are summarized in Table 1 in Appendix. We name these test suites as `{Robot}-{Constraint Type}-{Constraint Number}`.

**Comparison Group** The methods in the comparison group include: (i) unconstrained RL algorithm TRPO (Schulman et al., 2015) (ii) end-to-end constrained safe RL algorithms CPO (Achiam et al., 2017), TRPO-Lagrangian (Bohez et al., 2019), TRPO-FAC (Ma et al., 2021), TRPO-IPO (Liu et al., 2020), PCPO (Yang et al., 2020b), and (iii) hierarchical safe RL algorithms TRPO-SL (TRPO-Safety Layer) (Dalal et al., 2018), TRPO-USL (TRPO-Unrolling Safety Layer) (Zhang et al., 2022a). We select TRPO as our baseline method since it is state-of-the-art and already has safety-constrained derivatives that can be tested off-the-shelf. For hierarchical safe RL algorithms, we employ a warm-up phase (1/3 of the whole epochs) which does unconstrained TRPO training, and the generated data will be used to pre-train the safety critic for future epochs. For all experiments, the policy $\pi$, the value ($V^\pi, V^\pi_D$) are all encoded in feedforward neural networks using two hidden layers of size (64,64) with tanh activations. More details are provided in Appendix F.

**Evaluation Metrics** For comparison, we evaluate algorithm performance based on (i) **reward performance**, (ii) **average episode cost** and (iii) **cost rate** (state-wise cost). Comparison metric details are provided in Appendix F.3. We set the limit of cost to 0 for all the safe RL algorithms since we aim to avoid any violation of the constraints. For our comparison, we implement the baseline safe RL algorithms exactly following the policy update / action correction procedure from the original papers. We emphasize that in order for the comparison to be fair, we give baseline safe RL algorithms every advantage that is given to SCPO, including equivalent trust region policy updates.

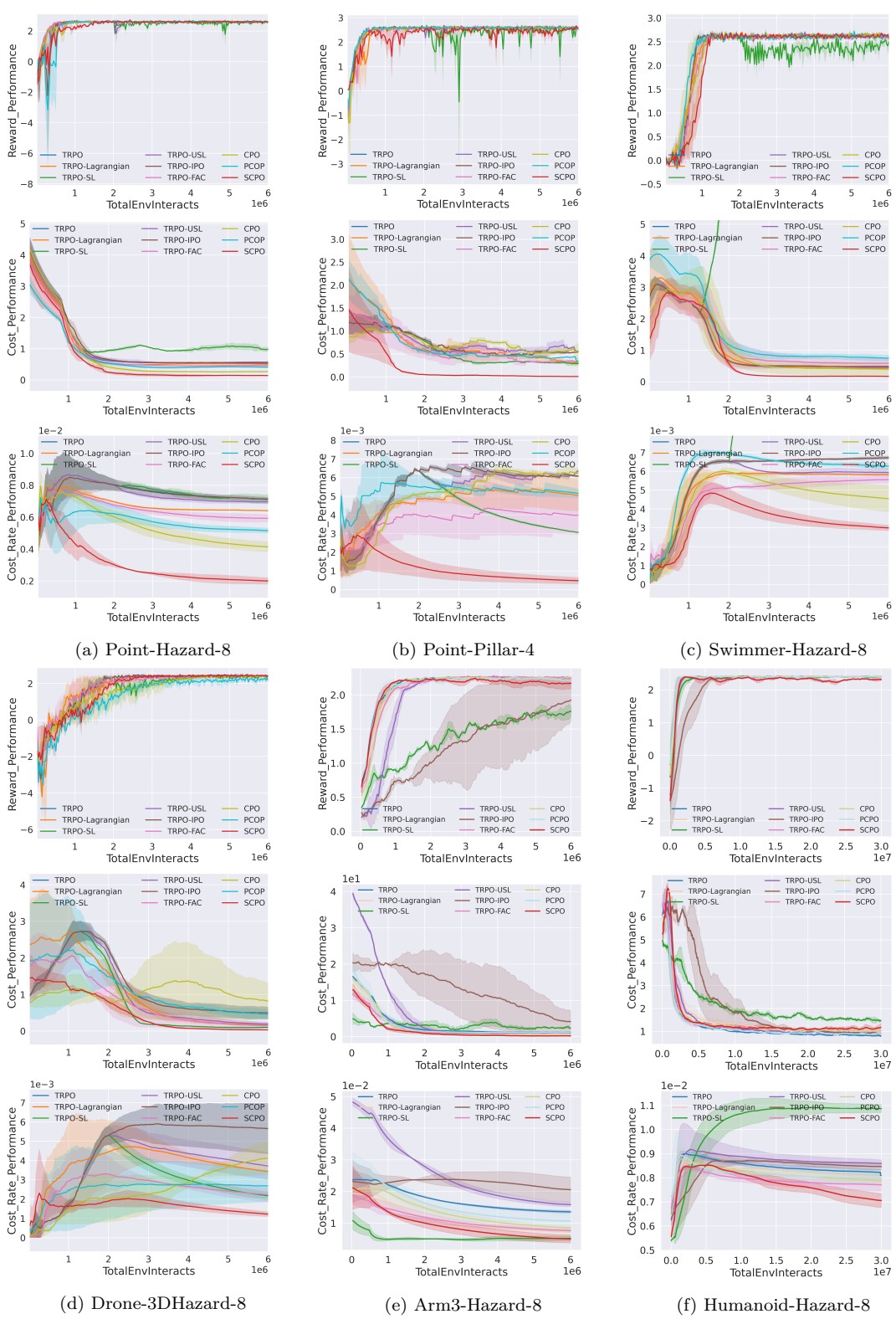

Figure 5: Comparison of results from (i) four representative test suites in low dimensional systems (Point, Swimmer, Drone), (ii) Arm reaching, and (iii) Humanoid locomotion.

## 6.2 Evaluating SCPO and Comparison Analysis

**Low Dimension System** We select four representative test suites on low dimensional system (Point, Swimmer, Drone) and summarize the comparison results on Figure 5, which demonstrate that SCPO is successful at approximately enforcing zero constraints violation safety performance in all environments after the policy converges. Specifically, compared with the baseline safe RL methods, SCPO is able to achieve (i) near zero average episode cost and (ii) significantly lower cost rate without sacrificing reward performance. The baseline end-to-end safe RL methods (TRPO-Lagrangian, TRPO-FAC, TRPO-IPO, CPO, PCPO) fail to achieve the near zero cost performance even when the cost limit is set to be 0. The baseline hierarchical safe RL methods (TRPO-SL, TRPO-USL) also fail to achieve near zero cost performance even with an explicit safety layer

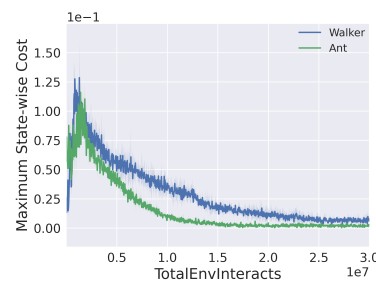

Figure 6:
Maximum state-wise cost

to correct the unsafe action at every time step. End-to-end safe RL algorithms fail since all methods rely on CMDP to minimize the discounted cumulative cost while SCPO directly work with MMDP to restrict the state-wise maximum cost by Proposition 1. We also observe that TRPO-SL fails to lower the violation during training, due to the fact that the linear approximation of cost function $C(\hat{s}_t, a, \hat{s}_{t+1})$ (Dalal et al., 2018) becomes inaccurate when the dynamics are highly nonlinear like the ones we used in MuJoCo (Todorov et al., 2012). More detailed metrics for comparison and experimental results on test suites with low dimension systems are summarized in Appendix F.3.

**High Dimension System** To demonstrate the scalability and performance of SCPO in high-dimensional systems, we conducted tests on the Ant-Hazard-8 Walker-Hazard-8 suites, with 8-dimensional and 10-dimensional control spaces, respectively. The comparison results for high-dimensional systems are summarized in Figure 2, which show that SCPO outperforms all other baselines in enforcing zero safety violation without compromising performance in terms of return. SCPO rapidly stabilizes the cost return around zero and significantly reduces the cost rate, while the other baselines fail to converge to a policy with near-zero cost.

Furthermore, we tackled more scenarios involving robot arm goal reaching and humanoid locomotion. The comparative results for these tasks are detailed in Figure 5e and Figure 5f, respectively. Notably, SCPO consistently achieves the lowest cost rate (state-wise cost) in both assignments. It's essential to highlight an observation in the humanoid task: while the episodic cost is higher compared to several baseline methods, the cost rate is the most favorable. This discrepancy arises because SCPO intentionally takes more cautious paths around hazards to ensure safety, leading to an increased number of time steps per episode. Consequently, although the state-wise cost is minimized, the average episodic cost rises due to the longer average episodic horizon.

The comparison results of both low dimension and high dimension systems answer **Q1**.

**Maximum State-wise Cost** As pointed in Section 3.3, the underlying magic for enabling near-zero safety violation is to restrict the maximum state-wise cost to stay around zero. To have a better understanding of this pro-

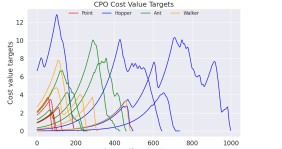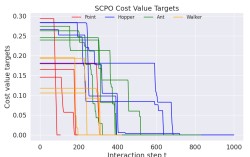

Figure 7: Cost value function target of five randomly sampled episode of different tasks

cess, we visualize the evolution of maximum state-wise cost for SCPO on the challenging high-dimensional Ant-Hazard-8 and Walker-Hazard-8 test suites in Figure 6 , which answers **Q2**. Within Figure 6, each data point is obtained by averaging the maximum state-wise cost across all episodes within the current epoch. To facilitate better comparisons with other works, we consistently employ the cost rate as a metric to illustrate the state-wise cost performance in our results.

**Ablation on Sub-sampling Imbalanced Cost Increment Value Targets** One critical step in SCPO involves learning the cost increments value functions $V_{D_i}^{\pi_k}(\hat{s}_t)$. These functions represent the maximum cost increment in any future state relative to the maximal state-wise cost encountered so far. In simpler terms, $V_{D_i}^{\pi_k}(\hat{s}_t)$ is a non-increasing step function. In particular, it resembles a staircase, with many steps locating at zero after

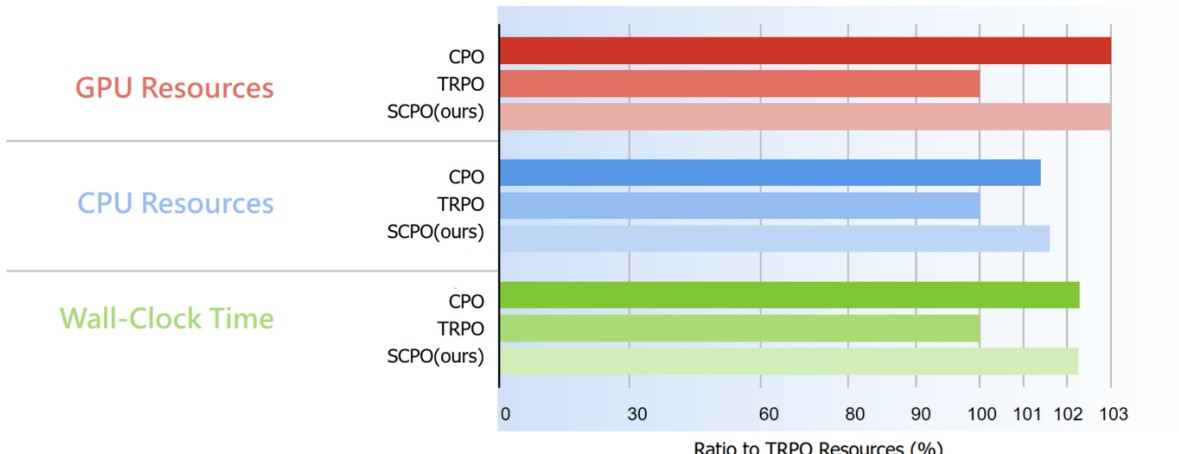

Figure 10: Resource usage compared to other algorithms under the Goal-Hazard-8 task, where TRPO is set as the baseline

the point where the maximum state-wise cost over the trajectory has been reached. This characteristic gives the cost increment value target a skewed distribution with a high frequency of zeros, as illustrated in Figure 7.

Sub-sampling is employed to bring down the zero targets population to match the population of non-zero targets. This adjustment is unique to SCPO, as other safe RL baselines try to fit a cost value function $V_{C_i'}^{\pi_k}(s_h) = \mathbb{E}_{\tau \sim \pi_k}\left[\sum_{t=h}^{H} \gamma^{t-h} C_i(s_t, a_t, s_{t+1})\right]$. In these baselines, the population of zero-cost value targets (only after encountering the last cost) is usually much smaller than the population of non-zero targets, rendering sub-sampling unnecessary. We also visualize $V_{C_i'}^{\pi_k}(s_h)$ targets in the left hand side of Figure 7. Consequently, applying sub-

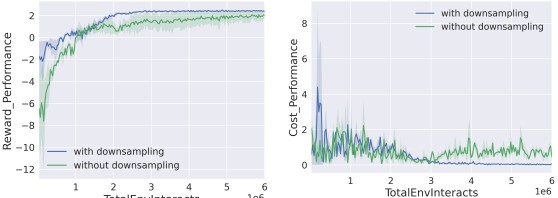

Figure 8: SCPO sub-sampling ablation study with Drone-3DHazard-8

sampling to reduce the zero $V_{C_i'}^{\pi_k}(\hat{s}_t)$ target population is irrelevant and has no impact on their performance.

Next, to demonstrate the necessity of sub-sampling for solving this challenge, we compare the performance of SCPO with and without sub-sampling trick on the aerial robot test suite, summarized in Figure 8. It is evident that with sub-sampling, the agent achieves higher rewards and more importantly, converges to near-zero costs. That is because sub-sampling effectively balances the cost increment value targets and improves the fitting of $V_{D_i}^{\pi_k}(\hat{s}_t)$. We also attempted to solve the imbalance issue via over-sampling non-zero targets, but did not observe promising results. This ablation study provides insights into **Q3**.

**Tightness of State-wise Maximum Cost Bound** To assess the tightness of the state-wise maximum cost bound in equation 12 (surrogate

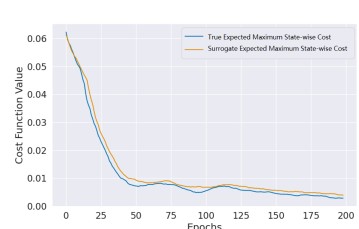

Figure 9: Visualization of true cost function and surrogate function

cost function in equation 11), we track and visualize the true values and surrogate values (bound estimates using the policy from the previous iteration) of $\mathcal{J}_D$ throughout policy training in the Point-Hazard-8 testing suite, as depicted in Figure 9. Due to approximation errors, a minor overlap between true values and surrogate values is noticeable at the initial stages of training. However, this overlap is quickly resolved, and thereafter, the surrogate cost consistently functions as a precise upper bound for the true value. The deviation, approximately 1e-3, is remarkable considering the scale of the true value, which ranges from 1e-2 to 5e-2. This observation underscores the good tightness in the theoretical state-wise maximum cost bound and answers **Q4**.

**Resources Usage Situation**   We conduct tests comparing GPU and CPU memory usage, as well as wall-clock time, for CPO, TRPO, and SCPO, all allocated with identical system resources in the Goal-Hazard-8 task. As illustrated in Figure 10, it is observed that CPO and SCPO exhibit nearly identical GPU occupancy and time consumption, with SCPO utilizing slightly more CPU resources. Notably, when considering the scale change on the horizontal axis, the three algorithms demonstrate comparable performance across all three metrics. This suggests that our algorithm achieves improved performance without noticeable increased system load or consuming additional time, affirming its superior overall efficiency. The results provide answer to **Q5**.

## 7   Conclusion and Future Work

This paper proposed SCPO, the first general-purpose policy search algorithm for state-wise constrained RL. Our approach provides guarantees for state-wise constraint satisfaction at each iteration, allows training of high-dimensional neural network policies while ensuring policy behavior, and is based on a new theoretical result on Maximum Markov decision process. We demonstrate SCPO's effectiveness on robot locomotion tasks, showing its significant performance improvement compared to existing methods and ability to handle state-wise constraints.

**Limitation and future work**   One limitation of our work is that, although SCPO satisfies state-wise constraints, the theoretical results are valid only in expectation, meaning that constraint violations are still possible during deployment. To address that, we will study absolute state-wise constraint satisfaction, i.e. bounding the *maximal possible* state-wise cost, which is even stronger than the current result (satisfaction in expectation).

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

# A    Complexity analysis for SCMDP

The complete form of equation 6 is:

$$\max_{\pi} \mathcal{J}_0(\pi), \text{ s.t. } \forall\big((s_t, a_t, s_{t+1}) \sim \tau, \tau \sim \pi\big), \ \forall i \in 1, \cdots, m, \ \mathbb{E}\big[C_i(s_t, a_t, s_{t+1})\big] \le w_i, \tag{17}$$

where each state-action transition pair corresponds to a constraint. Consider there's only one constraint function $C_1$, equation 17 is transformed as:

$$
\begin{aligned}
\max_{\pi} \ & \mathcal{J}_0(\pi), \\
\text{s.t.} \quad & \mathcal{G}_1(\pi) \doteq \mathop{\mathbb{E}}_{\substack{(s_0,a_0,s_1)\sim\tau \\ \tau\sim\pi}} \big[C_1(s_0, a_0, s_1)\big] - w_1 \le 0 \\
& \mathcal{G}_2(\pi) \doteq \mathop{\mathbb{E}}_{\substack{(s_1,a_1,s_2)\sim\tau \\ \tau\sim\pi}} \big[C_1(s_1, a_1, s_2)\big] - w_1 \le 0 \\
& \vdots \\
& \mathcal{G}_H(\pi) \doteq \mathop{\mathbb{E}}_{\substack{(s_{H-1},a_{H-1},s_H)\sim\tau \\ \tau\sim\pi}} \big[C_1(s_{H-1}, a_{H-1}, s_H)\big] - w_1 \le 0 \quad .
\end{aligned}
\tag{18}
$$

Suppose $\pi$ is parameterized by $\hat{\theta} \in \mathbb{R}^{n_\pi}$, With KKT conditions, equation 18 can be optimized via solving the following program:

$$
\begin{cases}
\frac{\partial \mathcal{L}(\pi,\lambda)}{\partial \pi_i} = 0, i = 1, 2, \cdots, n_\pi \\
\lambda_j \mathcal{G}_j(\pi) = 0, j = 1, 2, \cdots, H \\
\lambda_i \ge 0, j = 1, 2, \cdots, H \quad ,
\end{cases}
\tag{19}
$$

where $\mathcal{L}(\pi, \lambda) = \mathcal{J}_0(\pi) + \sum_{i=1}^{H} \lambda_i \mathcal{G}_i(\pi)$.

To understand the time complexity of equation 19, we can treat $\mathcal{J}_0$ and $\mathcal{G}_i$ as linear functions with respect to $\pi$. So that equation 19 represents a linear program, which can be solved by the fastest algorithm (Cohen et al., 2021) in time

$$O^*\big(((n_\pi + H)^\omega + (n_\pi + H)^{2.5-\alpha/2} + (n_\pi + H)^{2+1/6}) \log((n_\pi + H)/\delta)\big) \tag{20}$$

where $\omega$ is the exponent of matrix multiplicatoin, $\alpha$ is the dual exponent of matrix multiplication, and $\delta$ is the relative accuracy. For the current value of $\omega \sim 2.37$ and $\alpha \sim 0.31$, the state-of-the-art algorithm takes $O^*\big((n_\pi + H)^\omega \log((n_\pi + H)/\delta)\big)$ time (Cohen et al., 2021).

Consider (i) there are multiple cost functions $C_i$, and (ii) $\mathcal{J}_0$ and $\mathcal{G}_i$ are nonlinear functions with respect to $\pi$, the complexity of solving equation 17 with good accuracy, i.e. solving SCMDP, will be larger than $O^*\big((n_\pi + H)^2\big)$.

# B    Preliminaries

NEW

To facilitate the proof of Theorem 1, we begin by establishing key preliminaries that underpin the foundations of finite-horizon variations of the performance improvement bound, considering the discounted sum nature. The subsequent section, Appendix C, will elucidate the policy improvement of finite-horizon undiscounted sum Markov decision process (MDP). Our preliminary groundwork draws inspiration from [Appendix 10.1, Achiam et al. (2017)], extending the theoretical framework for finite-horizon scenarios.

$\dot{d}^\pi$ we used is defined as

$$\dot{d}^\pi(\hat{s}) = \sum_{t=0}^{H} \gamma^t P(\hat{s}_t = \hat{s}|\pi). \tag{21}$$

which has a little difference with $d^\pi$ and is used to ensure the continuity of function we used for proof later. Then it allows us to express the expected discounted total reward or cost compactly as:

$$\mathcal{J}_g(\pi) = \mathop{\mathrm{E}}_{\substack{\hat{s}\sim\dot{d}^\pi \\ a\sim\pi \\ \hat{s}'\sim P}} [g(\hat{s},a,\hat{s}')], \tag{22}$$

where by $a \sim \pi$, we mean $a \sim \pi(\cdot|\hat{s})$, and by $\hat{s}' \sim P$, we mean $\hat{s}' \sim P(\cdot|\hat{s},a)$. $g(\hat{s},a,\hat{s}')$ represents the cost or reward function. We drop the explicit notation for the sake of reducing clutter, but it should be clear from context that $a$ and $\hat{s}'$ depend on $\hat{s}$.

Define $P(\hat{s}'|\hat{s},a)$ is the probability of transitioning to state $\hat{s}'$ given that the previous state was $\hat{s}$ and the agent took action $a$ at state $\hat{s}$, and $\hat{\mu} : \hat{\mathcal{S}} \mapsto [0,1]$ is the initial augmented state distribution. Let $p_\pi^t \in \mathbb{R}^{|\hat{\mathcal{S}}|}$ denote the vector with components $p_\pi^t(\hat{s}) = P(\hat{s}_t = \hat{s}|\pi)$, and let $P_\pi \in \mathbb{R}^{|\hat{\mathcal{S}}|\times|\hat{\mathcal{S}}|}$ denote the transition matrix with components $P_\pi(\hat{s}'|\hat{s}) = \int P(\hat{s}'|\hat{s},a)\pi(a|\hat{s})da$; then $p_\pi^t = P_\pi p_\pi^{t-1} = P_\pi^t \hat{\mu}$ and

$$\begin{aligned} \dot{d}^\pi &= \sum_{t=0}^{H} (\gamma P_\pi)^t \hat{\mu} \\ &= (I - (\gamma P_\pi)^{H+1})(I - \gamma P_\pi)^{-1}\hat{\mu} \\ &= (I - \gamma P_\pi)^{-1}\hat{\mu} \end{aligned} \tag{23}$$

Noticing that the finite MDP ends up at step $H$, thus $(P_\pi)^{H+1}$ should be set to zero matrix.

This formulation helps us easily obtain the following lemma.

**Lemma 1.** *For any function $f : \hat{\mathcal{S}} \mapsto \mathbb{R}$ and any policy $\pi$,*

$$\mathop{\mathrm{E}}_{\hat{s}\sim\hat{\mu}}[f(\hat{s})] + \mathop{\mathrm{E}}_{\substack{\hat{s}\sim\dot{d}^\pi \\ a\sim\pi \\ \hat{s}'\sim P}}[\gamma f(\hat{s}')] - \mathop{\mathrm{E}}_{\hat{s}\sim\dot{d}^\pi}[f(\hat{s})] = 0. \tag{24}$$

*Proof.* Multiply both sides of equation 23 by $(I - \gamma P_\pi)$ and take the inner product with the vector $f \in \mathbb{R}^{|\hat{\mathcal{S}}|}$. □

Combining Lemma 1 with equation 22, we obtain the following, for any function $f$ and any policy $\pi$:

$$\mathcal{J}_g(\pi) = \mathop{\mathrm{E}}_{\hat{s}\sim\hat{\mu}}[f(\hat{s})] + \mathop{\mathrm{E}}_{\substack{\hat{s}\sim\dot{d}^\pi \\ a\sim\pi \\ \hat{s}'\sim P}}[g(\hat{s},a,\hat{s}') + \gamma f(\hat{s}') - f(\hat{s})] \tag{25}$$

**Lemma 2.** *For any function $f \mapsto \mathbb{R}$ and any policies $\pi'$ and $\pi$, define*

$$L_{\pi,f}(\pi') \doteq \mathop{\mathrm{E}}_{\substack{\hat{s}\sim\dot{d}^\pi \\ a\sim\pi \\ \hat{s}'\sim P}} \left[ \left( \frac{\pi'(a|\hat{s})}{\pi(a|\hat{s})} - 1 \right) (g(\hat{s},a,\hat{s}') + \gamma f(\hat{s}') - f(\hat{s})) \right], \tag{26}$$

*and $\epsilon_f^{\pi'} \doteq \max_{\hat{s}} |\mathrm{E}_{a\sim\pi',\hat{s}'\sim P}[R(\hat{s},a,\hat{s}') + \gamma f(\hat{s}') - f(\hat{s})]|$. Then the following bounds hold:*

$$\mathcal{J}_g(\pi') - \mathcal{J}_g(\pi) \geq L_{\pi,f}(\pi') - \epsilon_f^{\pi'} \left\| \dot{d}^{\pi'} - \dot{d}^\pi \right\|_1, \tag{27}$$

$$\mathcal{J}_g(\pi') - \mathcal{J}_g(\pi) \leq L_{\pi,f}(\pi') + \epsilon_f^{\pi'} \left\| \dot{d}^{\pi'} - \dot{d}^\pi \right\|_1, \tag{28}$$

*where $D_{TV}$ is the total variational divergence. Furthermore, the bounds are tight(when $\pi' = \pi$, the LHS and RHS are identically zero).*

*Proof.* First, for notational convenience, let $\delta_f(\hat{s}, a, \hat{s}') \doteq g(\hat{s}, a, \hat{s}') + \gamma f(\hat{s}') - f(\hat{s})$. By equation 25, we obtain the identity

$$\mathcal{J}_g(\pi') - \mathcal{J}_g(\pi) = \underset{\substack{\hat{s} \sim \dot{d}^{\pi'} \\ a \sim \pi' \\ \hat{s}' \sim P}}{E}[\delta_f(\hat{s}, a, \hat{s}')] - \underset{\substack{\hat{s} \sim \dot{d}^{\pi} \\ a \sim \pi \\ \hat{s}' \sim P}}{E}[\delta_f(\hat{s}, a, \hat{s}')] \tag{29}$$

Now, we restrict our attention to the first term in equation 29. Let $\dagger\delta_f^{\pi'} \in \mathbb{R}^{|\hat{\mathcal{S}}|}$ denote the vector of components, where $\dagger\delta_f^{\pi'}(\hat{s}) = \mathbb{E}_{a \sim \pi', \hat{s}' \sim P}[\delta_f(\hat{s}, a, \hat{s}')|\hat{s}]$. Observe that

$$\underset{\substack{\hat{s} \sim \dot{d}^{\pi'} \\ a \sim \pi' \\ \hat{s}' \sim P}}{E}[\delta_f(\hat{s}, a, \hat{s}')] = \left\langle \dot{d}^{\pi'}, \dagger\delta_f^{\pi'} \right\rangle$$

$$= \left\langle \dot{d}^{\pi}, \dagger\delta_f^{\pi'} \right\rangle + \left\langle \dot{d}^{\pi'} - \dot{d}^{\pi}, \dagger\delta_f^{\pi'} \right\rangle$$

With the Hölder's inequality; for any $p, q \in [1, \infty]$ such that $\frac{1}{p} + \frac{1}{q} = 1$, we have

$$\left\langle \dot{d}^{\pi}, \dagger\delta_f^{\pi'} \right\rangle + \left\| \dot{d}^{\pi'} - \dot{d}^{\pi} \right\|_p \left\| \dagger\delta_f^{\pi'} \right\|_q \geq \underset{\substack{\hat{s} \sim \dot{d}^{\pi'} \\ a \sim \pi' \\ \hat{s}' \sim P}}{E}[\delta_f(\hat{s}, a, \hat{s}')] \geq \left\langle \dot{d}^{\pi}, \dagger\delta_f^{\pi'} \right\rangle - \left\| d^{\pi'} - \dot{d}^{\pi} \right\|_p \left\| \dagger\delta_f^{\pi'} \right\|_q \tag{30}$$

We choose $p = 1$ and $q = \infty$; With $\left\| \dagger\delta_f^{\pi'} \right\|_\infty = \epsilon_f^{\pi'}$, and by the importance sampling identity, we have

$$\left\langle \dot{d}^{\pi}, \dagger\delta_f^{\pi'} \right\rangle = \underset{\substack{\hat{s} \sim \dot{d}^{\pi} \\ a \sim \pi' \\ \hat{s}' \sim P}}{E}[\delta_f(\hat{s}, a, \hat{s}')] \tag{31}$$

$$= \underset{\substack{\hat{s} \sim \dot{d}^{\pi} \\ a \sim \pi \\ \hat{s}' \sim P}}{E}\left[ \left( \frac{\pi'(a|\hat{s})}{\pi(a|\hat{s})} \right) \delta_f(\hat{s}, a, \hat{s}') \right]$$

After bringing equation 31, $\left\| \dagger\delta_f^{\pi'} \right\|_\infty$ into equation 30, then substract $\underset{\substack{\hat{s} \sim \dot{d}^{\pi} \\ a \sim \pi \\ \hat{s}' \sim P}}{E}[\delta_f(\hat{s}, a, \hat{s}')]$, the bounds are obtained.

The lower bound leads to equation 27, and the upper bound leads to equation 28. $\qquad\square$

**Lemma 3.** *The divergence between discounted future state visitation distributions, $\|\dot{d}^{\pi'} - \dot{d}^{\pi}\|_1$, is bounded by an average divergence of the policies $\pi'$ and $\pi$:*

$$\|\dot{d}^{\pi'} - \dot{d}^{\pi}\|_1 \leq 2 \sum_{t=0}^{H} \gamma^{t+1} \underset{\hat{s} \sim \dot{d}^{\pi}}{E}[D_{TV}(\pi'||\pi)[\hat{s}]], \tag{32}$$

*where $D_{TV}(\pi'||\pi)[\hat{s}] = \frac{1}{2} \sum_a |\pi'(a|\hat{s}) - \pi(a|\hat{s})|$.*

*Proof.* Firstly, we introduce an identity for the vector difference of the discounted future state visitation distributions on two different policies, $\pi'$ and $\pi$. Define the matrices $G \doteq (I - \gamma P_\pi)^{-1}, \bar{G} \doteq (I - \gamma P_{\pi'})^{-1}$, and $\Delta = P_{\pi'} - P_\pi$. Then:

$$G^{-1} - \bar{G}^{-1} = (I - \gamma P_\pi) - (I - \gamma P_{\pi'}) \tag{33}$$
$$= \gamma\Delta,$$

left-multiplying by $G$ and right-multiplying by $\bar{G}$, we obtain

$$\bar{G} - G = \gamma \bar{G} \Delta G. \tag{34}$$

Thus, the following equality holds:

$$
\begin{aligned}
\dot{d}^{\pi'} - \dot{d}^{\pi} &= (1 - \gamma) \left( \bar{G} - G \right) \hat{\mu} \\
&= \gamma(1 - \gamma) \bar{G} \Delta G \hat{\mu} \\
&= \gamma \bar{G} \Delta \dot{d}^{\pi}.
\end{aligned} \tag{35}
$$

Using equation 35, we obtain

$$
\begin{aligned}
\|\dot{d}^{\pi'} - \dot{d}^{\pi}\|_1 &= \gamma \|\bar{G} \Delta \dot{d}^{\pi}\|_1 \\
&\leq \gamma \|\bar{G}\|_1 \|\Delta \dot{d}^{\pi}\|_1,
\end{aligned} \tag{36}
$$

where $\|\bar{G}\|_1$ is bounded by:

$$\|\bar{G}\|_1 = \|(I - \gamma P_{\pi'})^{-1}\|_1 \leq \sum_{t=0}^{\infty} \gamma^t \|P_{\pi'}^t\|_1 = \sum_{t=0}^{H} \gamma^t. \tag{37}$$

Next, we bound $\|\Delta \dot{d}_1^{\pi}\|$ as following:

$$
\begin{aligned}
\|\Delta \dot{d}^{\pi}\|_1 &= \sum_{\hat{s}'} \left| \sum_{\hat{s}} \Delta(\hat{s}'|\hat{s}) \dot{d}^{\pi}(\hat{s}) \right| \\
&\leq \sum_{\hat{s},\hat{s}'} |\Delta(\hat{s}'|\hat{s})| \dot{d}^{\pi}(\hat{s}) \\
&= \sum_{\hat{s},\hat{s}'} \left| \sum_{a} P(\hat{s}'|\hat{s}, a) \left( \pi'(a|\hat{s}) - \pi(a|\hat{s}) \right) \right| \dot{d}^{\pi}(\hat{s}) \\
&\leq \sum_{\hat{s},a,\hat{s}'} P(\hat{s}'|\hat{s}, a) |\pi'(a|\hat{s}) - \pi(a|\hat{s})| \dot{d}^{\pi}(\hat{s}) \\
&= \sum_{\hat{s},a} |\pi'(a|\hat{s}) - \pi(a|\hat{s})| \dot{d}^{\pi}(\hat{s}) \\
&= 2 \operatorname*{E}_{\hat{s} \sim \dot{d}^{\pi}} [D_{TV}(\pi'||\pi)[\hat{s}]].
\end{aligned} \tag{38}
$$

By taking equation 38 and equation 37 into equation 36, this lemma is proved.

□

The new policy improvement bound follows immediately.

**Lemma 4.** *For any function* $f : \hat{\mathcal{S}} \mapsto \mathbb{R}$ *and any policies* $\pi'$ *and* $\pi$, *define* $\delta_f(\hat{s}, a, \hat{s}') \doteq g(\hat{s}, a, \hat{s}') + \gamma f(\hat{s}') - f(\hat{s})$,

$$\epsilon_f^{\pi'} \doteq \max_{\hat{s}} |\mathrm{E}_{a \sim \pi', \hat{s}' \sim P}[\delta_f(\hat{s}, a, \hat{s}')]|,$$

$$L_{\pi,f}(\pi') \doteq \mathop{\mathrm{E}}_{\substack{\hat{s} \sim \hat{d}^\pi \\ a \sim \pi \\ \hat{s}' \sim P}} \left[ \left( \frac{\pi'(a|\hat{s})}{\pi(a|\hat{s})} - 1 \right) \delta_f(\hat{s}, a, \hat{s}') \right], and$$

$$D_{\pi,f}^{\pm}(\pi') \doteq L_{\pi,f}(\pi') \pm 2(\sum_{t=0}^{H} \gamma^{t+1}) \epsilon_f^{\pi'} \mathop{\mathrm{E}}_{\hat{s} \sim \hat{d}^\pi}[D_{TV}(\pi'||\pi)[\hat{s}]],$$

*where* $D_{TV}(\pi'||\pi)[\hat{s}] = \frac{1}{2} \sum_a |\pi'(a|\hat{s}) - \pi(a|\hat{s})|$ *is the total variational divergence between action distributions at* $\hat{s}$. *The following bounds hold:*

$$D_{\pi,f}^{+}(\pi') \geq \mathcal{J}_g(\pi') - \mathcal{J}_g(\pi) \geq D_{\pi,f}^{-}(\pi').$$

*Furthermore, the bounds are tight (when* $\pi' = \pi$, *all three expressions are identically zero)*

*Proof.* Begin with the bounds from lemma 2 and bound the divergence $D_{TV}(\hat{d}^{\pi'}||\hat{d}^\pi)$ by lemma 3. $\qquad\square$

## C  Proof for Theorem 1

*Proof.* The choice of $f = \hat{V}_D^\pi, g = D$ in lemma 4 leads to following inequality:

$$\hat{\mathcal{J}}_D(\pi') - \hat{\mathcal{J}}_D(\pi) \leq \mathop{\mathbb{E}}_{\substack{\hat{s} \sim \hat{d}^\pi \\ a \sim \pi'}} \left[ \hat{A}_D^\pi(\hat{s}, a) + 2(\sum_{t=0}^{H} \gamma^{t+1}) \epsilon_D^{\pi'} \mathcal{D}_{TV}(\pi'||\pi)[\hat{s}] \right]. \tag{39}$$

where $\hat{\mathcal{J}}_D(\pi) = \mathbb{E}_{\tau \sim \pi} \left[ \sum_{t=0}^{H} \gamma^t D(\hat{s}_t, a_t, \hat{s}_{t+1}) \right]$, need to distinguish from $\mathcal{J}_D(\pi)$. And $\hat{V}_D^\pi, \hat{A}_D^\pi$ are also the discounted version of $V_D^\pi$ and $A_D^\pi$. Note that according to Lemma 4 one can only get this the inequality holds when $\gamma \in (0, 1)$.

Then we can define $\mathcal{F}(\gamma) = \mathop{\mathbb{E}}_{\substack{\hat{s} \sim \hat{d}^\pi \\ a \sim \pi'}} \left[ \hat{A}_D^\pi(\hat{s}, a) + 2(\sum_{t=0}^{H} \gamma^{t+1}) \epsilon_D^{\pi'} \mathcal{D}_{TV}(\pi'||\pi)[\hat{s}] \right] - \hat{\mathcal{J}}_D(\pi') + \hat{\mathcal{J}}_D(\pi)$ with the following condition holds:

$$\mathcal{F}(\gamma) \geq 0, \text{when } \gamma \in (0, 1) \tag{40}$$
$$\mathcal{F}(\gamma)\text{'s domain of definition is } \mathcal{R}$$
$$\mathcal{F}(\gamma) \text{ is a polynomial function}$$

Because $\mathcal{F}(\gamma)$ is a polynomial function and coefficients are all limited, thus $\lim_{\gamma \to 1^-} \mathcal{F}(\gamma)$ exists and $\mathcal{F}(\gamma)$ is continuous at point $(1, \mathcal{F}(1))$. So $\mathcal{F}(1) = \lim_{\gamma \to 1^-} \mathcal{F}(\gamma) \geq 0$, which equals to:

$$\mathcal{J}_D(\pi') - \mathcal{J}_D(\pi) \leq \mathop{\mathbb{E}}_{\substack{\hat{s} \sim \bar{d}^\pi \\ a \sim \pi'}} \left[ A_D^\pi(\hat{s}, a) + 2(H+1) \epsilon_D^{\pi'} \mathcal{D}_{TV}(\pi'||\pi)[\hat{s}] \right].$$

where $\bar{d}^\pi = \sum_{t=0}^{H} P(\hat{s}_t = \hat{s}|\pi)$.

$\qquad\square$

# D   Proof for Proposition 2

*Proof.* Here we first present a new bound on the difference in returns between two arbitrary policies in the context of finite-horizon MDP:

**Theorem 2** (Trust Region Update Performance). *For any policies $\pi', \pi$, with $\epsilon^{\pi'} \doteq max_{\hat{s}}|E_{a \sim \pi'}[A^{\pi}(\hat{s}, a)]|$, and define $d^{\pi} = (1 - \gamma) \sum_{t=0}^{H} \gamma^t P(\hat{s}_t = \hat{s}|\pi)$ as the discounted augmented state distribution using $\pi$, then the following bound holds:*

$$\mathcal{J}(\pi') - \mathcal{J}(\pi) > \frac{1}{1-\gamma} \underset{\substack{\hat{s} \sim d^{\pi} \\ a \sim \pi'}}{\mathbb{E}} \left[ A^{\pi}(\hat{s}, a) - \frac{2\gamma\epsilon^{\pi'}}{1-\gamma} \mathcal{D}_{TV}(\pi' \| \pi)[\hat{s}] \right] \tag{41}$$

We provide the proof for Theorem 2 in Appendix D.2. The following bound then follows directly from Theorem 2 using the relationship between the total variation divergence and the KL divergence:

$$\mathcal{J}(\pi') - \mathcal{J}(\pi) > \frac{1}{1-\gamma} \underset{\substack{\hat{s} \sim d^{\pi} \\ a \sim \pi'}}{\mathbb{E}} \left[ A^{\pi}(\hat{s}, a) - \frac{2\gamma\epsilon^{\pi'}}{1-\gamma} \sqrt{\frac{1}{2} \mathbb{E}_{\hat{s} \sim d^{\pi}} [\mathcal{D}_{KL}(\pi' \| \pi)[\hat{s}]]} \right]. \tag{42}$$

In equation 11, the reward performance between two policies is associated with trust region, i.e.

$$\pi_{k+1} = \underset{\pi \in \Pi_{\theta}}{\textbf{argmax}} \underset{\substack{\hat{s} \sim d^{\pi_k} \\ a \sim \pi}}{\mathbb{E}} [A^{\pi_k}(\hat{s}, a)] \tag{43}$$
$$\textbf{s.t.} \quad \mathbb{E}_{\hat{s} \sim \bar{d}^{\pi_k}} [\mathcal{D}_{KL}(\pi \| \pi_k)[\hat{s}]] \le \delta.$$

Due to Lemma 5 (proved in Appendix D.1), if two policies are related with Equation (43), they are related with the following optimization:

$$\pi_{k+1} = \underset{\pi \in \Pi_{\theta}}{\textbf{argmax}} \underset{\substack{\hat{s} \sim d^{\pi_k} \\ a \sim \pi}}{\mathbb{E}} [A^{\pi_k}(\hat{s}, a)] \tag{44}$$
$$\textbf{s.t.} \quad \mathbb{E}_{\hat{s} \sim d^{\pi_k}} [\mathcal{D}_{KL}(\pi \| \pi_k)[\hat{s}]] \le \delta.$$

By equation 42 and equation 44, if $\pi_k, \pi_{k+1}$ are related by equation 11, then performance return for $\pi_{k+1}$ satisfies

$$\mathcal{J}(\pi_{k+1}) - \mathcal{J}(\pi_k) > -\frac{\sqrt{2\delta}\gamma\epsilon^{\pi_{k+1}}}{1-\gamma}.$$

$\square$

## D.1   KL Divergence Relationship Between $d^{\pi_k}$ and $\bar{d}^{\pi_k}$

**Lemma 5.** $\underset{\hat{s} \sim d_{\pi}}{\text{E}} [\mathcal{D}_{KL}(\pi' \| \pi)[\hat{s}]] < \underset{\hat{s} \sim \bar{d}_{\pi}}{\text{E}} [\mathcal{D}_{KL}(\pi' \| \pi)[\hat{s}]]$

*Proof.*

$$\mathop{\mathrm{E}}_{\hat{s}\sim d_\pi}[\mathcal{D}_{KL}(\pi'\|\pi)[\hat{s}]] = \sum_{\hat{s}}(1-\gamma)\sum_{t=0}^{H}\gamma^t P(\hat{s}_t = \hat{s}|\pi)\mathcal{D}_{KL}(\pi'\|\pi)[\hat{s}]$$

$$< \sum_{\hat{s}}\sum_{t=0}^{H}\gamma^t P(\hat{s}_t = \hat{s}|\pi)\mathcal{D}_{KL}(\pi'\|\pi)[\hat{s}]$$

$$< \sum_{\hat{s}}\sum_{t=0}^{H}P(\hat{s}_t = \hat{s}|\pi)\mathcal{D}_{KL}(\pi'\|\pi)[\hat{s}]$$

$$= \mathop{\mathrm{E}}_{\hat{s}\sim \tilde{d}_\pi}[\mathcal{D}_{KL}(\pi'\|\pi)[\hat{s}]].$$

$\square$

### D.2 Proof for Theorem 2

The choice of $f = V_\pi, g = R$ in lemma 4 leads to following inequality:

For any policies $\pi', \pi$, with $\epsilon^{\pi'} \doteq max_{\hat{s}}|E_{a\sim\pi'}[A_\pi(\hat{s}, a)]|$, the following bound holds:

$$\mathcal{J}(\pi') - \mathcal{J}(\pi) \geq \mathop{\mathrm{E}}_{\substack{\hat{s}\sim \tilde{d}^\pi \\ a\sim\pi'}}\left[A_\pi(\hat{s}, a) - 2(\sum_{t=0}^{H}\gamma^{t+1})\epsilon^{\pi'}D_{TV}(\pi'\|\pi)[\hat{s}]\right]$$

$$> \frac{1}{1-\gamma}\mathop{\mathrm{E}}_{\substack{\hat{s}\sim d^\pi \\ a\sim\pi'}}\left[A_\pi(\hat{s}, a) - \frac{2\gamma\epsilon^{\pi'}}{1-\gamma}D_{TV}(\pi'\|\pi)[\hat{s}]\right]$$

At this point, the theorem 2 is proved.

# E  SCPO Pseudocode

---

**Algorithm 1** State-wise Constrained Policy Optimization

---

**Input:** Initial policy $\pi_0 \in \Pi_\theta$.

**for** $k = 0, 1, 2, \ldots$ **do**

    Sample trajectory $\tau \sim \pi_k = \pi_{\theta_k}$

    Estimate gradient $g \leftarrow \nabla_\theta \mathbb{E}_{\hat{s}, a \sim \tau}\left[A^\pi(\hat{s}, a)\right]\big|_{\theta=\theta_k}$                ▷ section 5

    Estimate gradient $b_i \leftarrow \nabla_\theta \mathbb{E}_{\hat{s}, a \sim \tau}\left[A^\pi_{D_i}(\hat{s}, a)\right]\big|_{\theta=\theta_k}, \ \forall i = 1, 2, \ldots, m$       ▷ section 5

    Estimate Hessian $H \leftarrow \nabla_\theta^2 \mathbb{E}_{\hat{s} \sim \tau}[\mathcal{D}_{KL}(\pi \| \pi_k)[\hat{s}]]\big|_{\theta=\theta_k}$

    Solve convex programming                             ▷ Achiam et al. (2017)

$$\theta_{k+1}^* = \underset{\theta}{\mathbf{argmax}} \quad g^\top(\theta - \theta_k)$$

$$\mathbf{s.t.} \quad \frac{1}{2}(\theta - \theta_k)^\top H(\theta - \theta_k) \le \delta$$

$$c_i + b_i^\top(\theta - \theta_k) \le 0, \ i = 1, 2, \ldots, m$$

    Get search direction $\Delta\theta^* \leftarrow \theta_{k+1}^* - \theta_k$

    **for** $j = 0, 1, 2, \ldots$ **do**                                           ▷ Line search

        $\theta' \leftarrow \theta_k + \xi^j \Delta\theta^*$                    ▷ $\xi \in (0, 1)$ is the backtracking coefficient

        **if** $\mathbb{E}_{\hat{s} \sim \tau}[\mathcal{D}_{KL}(\pi_{\theta'} \| \pi_k)[\hat{s}]] \le \delta$ **and**                      ▷ Trust region

           $\mathbb{E}_{\hat{s}, a \sim \tau}\left[A^{\pi_{\theta'}}_{D_i}(\hat{s}, a)\right] - \mathbb{E}_{\hat{s}, a \sim \tau}\left[A^{\pi_k}_{D_i}(\hat{s}, a)\right] \le \max(-c_i, 0), \ \forall i$ **and**        ▷ Costs

           $(\mathbb{E}_{\hat{s}, a \sim \tau}\left[A^{\pi_{\theta'}}(\hat{s}, a)\right] \ge \mathbb{E}_{\hat{s}, a \sim \tau}\left[A^{\pi_k}(\hat{s}, a)\right]$ **or** infeasible equation 11) **then**     ▷ Rewards

           $\theta_{k+1} \leftarrow \theta'$                                        ▷ Update policy

           **break**

        **end if**

    **end for**

**end for**

---

Table 1: The test suites environments of our experiments

| | Ground robot | | | | | | Aerial robot |
|---|---|---|---|---|---|---|---|
| **Task Setting** | Low dimension | | | High dimension | | | |
| | Point | Swimmer | Arm3 | Walker | Ant | Humanoid | Drone |
| Hazard-1 | ✓ | ✓ | | | | | |
| Hazard-4 | ✓ | ✓ | | | | | |
| Hazard-8 | ✓ | ✓ | ✓ | ✓ | ✓ | ✓ | |
| Pillar-1 | ✓ | | | | | | |
| Pillar-4 | ✓ | | | | | | |
| Pillar-8 | ✓ | | | | | | |
| 3DHazard-1 | | | | | | | ✓ |
| 3DHazard-4 | | | | | | | ✓ |
| 3DHazard-8 | | | | | | | ✓ |

## F  Expeiment Details

### F.1  Environment Settings

**Goal Task**   In the Goal task environments, the reward function is:

$$r(x_t) = d_{t-1}^g - d_t^g + \mathbf{1}[d_t^g < R^g] \ ,$$

where $d_t^g$ is the distance from the robot to its closest goal and $R^g$ is the size (radius) of the goal. When a goal is achieved, the goal location is randomly reset to someplace new while keeping the rest of the layout the same. The test suites of our experiments are summarized in Table 1.

**Hazard Constraint**   In the Hazard constraint environments, the cost function is:

$$c(x_t) = \max(0, R^h - d_t^h) \ ,$$

where $d_t^h$ is the distance to the closest hazard and $R^h$ is the size (radius) of the hazard.

**Pillar Constraint**   In the Pillar constraint environments, the cost $c_t = 1$ if the robot contacts with the pillar otherwise $c_t = 0$.

**State Space**   The state space is composed of two parts. The internal state spaces describe the state of the robots, which can be obtained from standard robot sensors (accelerometer, gyroscope, magnetometer, velocimeter, joint position sensor, joint velocity sensor and touch sensor). The details of the internal state spaces of the robots in our test suites are summarized in Table 2. The external state spaces are describe the state of the environment observed by the robots, which can be obtained from 2D lidar or 3D lidar (where each lidar sensor perceives objects of a single kind). The state spaces of all the test suites are summarized in Table 3. Note that Vase and Gremlin are two other constraints in Safety Gym (Ray et al., 2019) and all the returns of vase lidar and gremlin lidar are zero vectors (i.e., $[0, 0, \cdots, 0] \in \mathbb{R}^{16}$) in our experiments since none of our test suites environments has vases.

**Control Space**   For all the experiments, the control space of all robots are continuous, and linearly scaled to [-1, +1].

### F.2  Policy Settings

The hyper-parameters used in our experiments are listed in Table 4 as default.

Table 2: The internal state space components of different test suites environments.

| Internal State Space | Point | Swimmer | Walker | Ant | Drone | Arm3 | Humanoid |
|---|---|---|---|---|---|---|---|
| Accelerometer ($\mathbb{R}^3$) | ✓ | ✓ | ✓ | ✓ | ✓ | ✓×5 | ✓ |
| Gyroscope ($\mathbb{R}^3$) | ✓ | ✓ | ✓ | ✓ | ✓ | ✓×5 | ✓ |
| Magnetometer ($\mathbb{R}^3$) | ✓ | ✓ | ✓ | ✓ | ✓ | ✓×5 | ✓ |
| Velocimeter ($\mathbb{R}^3$) | ✓ | ✓ | ✓ | ✓ | ✓ | ✓×5 | ✓ |
| Joint position sensor ($\mathbb{R}^n$) | $n=0$ | $n=2$ | $n=10$ | $n=8$ | $n=0$ | $n=3$ | $n=17$ |
| Joint velocity sensor ($\mathbb{R}^n$) | $n=0$ | $n=2$ | $n=10$ | $n=8$ | $n=0$ | $n=3$ | $n=17$ |
| Touch sensor ($\mathbb{R}^n$) | $n=0$ | $n=4$ | $n=2$ | $n=8$ | $n=0$ | $n=1$ | $n=2$ |

Table 3: The external state space components of different test suites environments.

| External State Space | Goal-Hazard | 3D-Goal-Hazard | Goal-Pillar |
|---|---|---|---|
| Goal Compass ($\mathbb{R}^3$) | ✓ | ✓ | ✓ |
| Goal Lidar ($\mathbb{R}^{16}$) | ✓ | ✗ | ✓ |
| 3D Goal Lidar ($\mathbb{R}^{60}$) | ✗ | ✓ | ✗ |
| Hazard Lidar ($\mathbb{R}^{16}$) | ✓ | ✗ | ✗ |
| 3D Hazard Lidar ($\mathbb{R}^{60}$) | ✗ | ✓ | ✗ |
| Pillar Lidar ($\mathbb{R}^{16}$) | ✗ | ✗ | ✓ |
| Vase Lidar ($\mathbb{R}^{16}$) | ✓ | ✗ | ✓ |
| Gremlin Lidar ($\mathbb{R}^{16}$) | ✓ | ✗ | ✓ |

Our experiments use separate multi-layer perception with *tanh* activations for the policy network, value network and cost network. Each network consists of two hidden layers of size (64,64). All of the networks are trained using *Adam* optimizer with learning rate of 0.01.

We apply an on-policy framework in our experiments. During each epoch the agent interact $B$ times with the environment and then perform a policy update based on the experience collected from the current epoch. The maximum length of the trajectory is set to 1000 and the total epoch number $N$ is set to 200 as default. In our experiments the Walker and the Ant were trained for 1000 epochs due to the high dimension.

The policy update step is based on the scheme of TRPO, which performs up to 100 steps of backtracking with a coefficient of 0.8 for line searching.

For all experiments, we use a discount factor of $\gamma = 0.99$, an advantage discount factor $\lambda = 0.95$, and a KL-divergence step size of $\delta_{KL} = 0.02$.

For experiments which consider cost constraints we adopt a target cost $\delta_c = 0.0$ to pursue a zero-violation policy.

Other unique hyper-parameters for each algorithms are hand-tuned to attain reasonable performance.

Each model is trained on a server with a 48-core Intel(R) Xeon(R) Silver 4214 CPU @ 2.2.GHz, Nvidia RTX A4000 GPU with 16GB memory, and Ubuntu 20.04.

For low-dimensional tasks, we train each model for 6e6 steps which takes around seven hours. For high-dimensional tasks, we train each model for 3e7 steps which takes around 60 hours.

### F.3 Metrics Comparison

In Tables 5 to 9, we report all the 14 results of our test suites by three metrics:

Table 4: Important hyper-parameters of different algorithms in our experiments

| Policy Parameter | | TRPO | TRPO-Lagrangian | TRPO-SL [18' Dalal] | TRPO-USL | TRPO-IPO | TRPO-FAC | CPO | PCPO | SCPO |
|---|---|---|---|---|---|---|---|---|---|---|
| Epochs | $N$ | 200 | 200 | 200 | 200 | 200 | 200 | 200 | 200 | 200 |
| Steps per epoch | $B$ | 30000 | 30000 | 30000 | 30000 | 30000 | 30000 | 30000 | 30000 | 30000 |
| Maximum length of trajectory | $L$ | 1000 | 1000 | 1000 | 1000 | 1000 | 1000 | 1000 | 1000 | 1000 |
| Policy network hidden layers | | (64, 64) | (64, 64) | (64, 64) | (64, 64) | (64, 64) | (64, 64) | (64, 64) | (64, 64) | (64, 64) |
| Discount factor | $\gamma$ | 0.99 | 0.99 | 0.99 | 0.99 | 0.99 | 0.99 | 0.99 | 0.99 | 0.99 |
| Advantage discount factor | $\lambda$ | 0.97 | 0.97 | 0.97 | 0.97 | 0.97 | 0.97 | 0.97 | 0.97 | 0.97 |
| TRPO backtracking steps | | 100 | 100 | 100 | 100 | 100 | 100 | 100 | - | 100 |
| TRPO backtracking coefficient | | 0.8 | 0.8 | 0.8 | 0.8 | 0.8 | 0.8 | 0.8 | - | 0.8 |
| Target KL | $\delta_{KL}$ | 0.02 | 0.02 | 0.02 | 0.02 | 0.02 | 0.02 | 0.02 | 0.02 | 0.02 |
| Value network hidden layers | | (64, 64) | (64, 64) | (64, 64) | (64, 64) | (64, 64) | (64, 64) | (64, 64) | (64, 64) | (64, 64) |
| Value network iteration | | 80 | 80 | 80 | 80 | 80 | 80 | 80 | 80 | 80 |
| Value network optimizer | | Adam | Adam | Adam | Adam | Adam | Adam | Adam | Adam | Adam |
| Value learning rate | | 0.001 | 0.001 | 0.001 | 0.001 | 0.001 | 0.001 | 0.001 | 0.001 | 0.001 |
| Cost network hidden layers | | - | (64, 64) | (64, 64) | (64, 64) | - | (64, 64) | (64, 64) | (64, 64) | (64, 64) |
| Cost network iteration | | - | 80 | 80 | 80 | - | 80 | 80 | 80 | 80 |
| Cost network optimizer | | - | Adam | Adam | Adam | - | Adam | Adam | Adam | Adam |
| Cost learning rate | | - | 0.001 | 0.001 | 0.001 | - | 0.001 | 0.001 | 0.001 | 0.001 |
| Target Cost | $\delta_c$ | - | 0.0 | 0.0 | 0.0 | 0.0 | 0.0 | 0.0 | 0.0 | 0.0 |
| Lagrangian optimizer | | - | - | - | - | - | Adam | - | - | - |
| Lagrangian learning rate | | - | 0.005 | - | - | - | 0.0001 | - | - | - |
| USL correction iteration | | - | - | - | 20 | - | - | - | - | - |
| USL correction rate | | - | - | - | 0.05 | - | - | - | - | - |
| Warmup ratio | | - | - | 1/3 | 1/3 | - | - | - | - | - |
| IPO parameter | $t$ | - | - | - | - | 0.01 | - | - | - | - |
| Cost reduction | | - | - | - | - | - | - | 0.0 | - | 0.0 |

- The average episode return $J_r$.

- The average episodic sum of costs $M_c$.

- The average state-wise cost over the entirety of training $\rho_c$.

All of the three metrics were obtained from the final epoch after convergence. Each metric was averaged over two random seed.

The learning curves of all experiments are shown in Figures 11 to 15.

A few general trends can be observed:

- All methods can converge to good reward performance under different task settings after about $1e6$ time steps. However, it often takes more time for the cost performance to get converge.

- The reward learning speed and the cost learning rate trade off against each other because the algorithms without state-wise constraints are more likely to explore unsafe state to gather more rewards.

### F.4   Ablation study on large penalty for infractions

We used adaptive penalty coefficient in our experiments with the Lagrangian method. Thus, we scale it up by a certain amount $\lambda$ to perform an investigation of the balance between reward and satisfying constraints. We name the experiment TRPO-LAG-$\{\lambda\}$ and compare it with SCPO in Figure 16. We can see that the cost rate and cost value of the

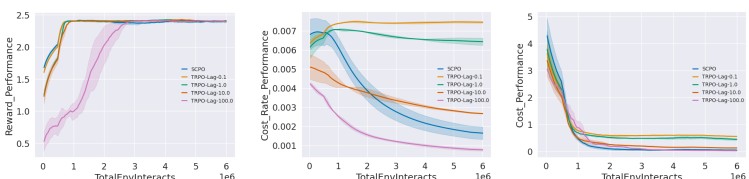

Figure 16: TRPO-Lagrangian method ablation study with Point-Hazard-8

Lagrangian method decreases significantly when lambda increases. But at the same time, the speed of convergence of the reward is greatly reduced. On the contrary, SCPO achieves the fastest convergence speed and the best convergence value in terms of both reward convergence and cost value decrease, and at the same time, it is not inferior in terms of cost rate. This shows that the simple coefficient adjustment of the Lagrangian method is not comparable to the superiority of our algorithm.

## G   Broader Impact

Our SCPO algorithm has been theoretically proven to effectively enforce state-wise instantaneous constraints, including safety-critical ones such as collision avoidance. However, achieving zero constraint violation in practical applications requires careful fine-tuning of the implementation and training process. Factors such as neural network structure, learning rate, and cost limits need to be properly adjusted to the specific task at hand. It is important to note that improper implementation and training of SCPO can still result in constraint violations, posing potential safety risks. Therefore, when deploying SCPO policies in safety-critical applications, it is strongly recommended to incorporate an explicit safety monitor, such as control saturation, to completely eliminate any potential safety issues.

Table 5: Metrics of three **Point-Hazard** environments obtained from the final epoch.

(a) Point-Hazard-1

| Algorithm | $\bar{J}_r$ | $\bar{M}_c$ | $\bar{\rho}_c$ |
|---|---|---|---|
| TRPO | 2.5779 | 0.7340 | 0.0086 |
| TRPO-Lagrangian | **2.6313** | 0.5977 | 0.0058 |
| TRPO-SL | 2.4721 | 11.7396 | 0.0116 |
| TRPO-USL | 2.5410 | 0.5381 | 0.0083 |
| TRPO-IPO | 2.5779 | 0.7340 | 0.0086 |
| TRPO-FAC | 2.5731 | 0.3263 | 0.0040 |
| CPO | 2.4988 | 0.1713 | 0.0045 |
| PCPO | 2.4928 | 0.3765 | 0.0054 |
| SCPO | 2.5822 | **0.0807** | **0.0013** |

(b) Point-Hazard-4

| Algorithm | $\bar{J}_r$ | $\bar{M}_c$ | $\bar{\rho}_c$ |
|---|---|---|---|
| TRPO | 2.5925 | 0.2412 | 0.0037 |
| TRPO-Lagrangian | 2.5494 | 0.2108 | 0.0034 |
| TRPO-SL | 2.5174 | 0.2915 | 0.0037 |
| TRPO-USL | **2.6140** | 0.2695 | 0.0035 |
| TRPO-IPO | 2.5946 | 0.2297 | 0.0038 |
| TRPO-FAC | 2.5566 | 0.1848 | 0.0028 |
| CPO | 2.5924 | 0.1654 | 0.0024 |
| PCPO | 2.5575 | 0.1824 | 0.0025 |
| SCPO | 2.5607 | **0.0687** | **0.0009** |

(c) Point-Hazard-8

| Algorithm | $\bar{J}_r$ | $\bar{M}_c$ | $\bar{\rho}_c$ |
|---|---|---|---|
| TRPO | 2.5761 | 0.5413 | 0.0071 |
| TRPO-Lagrangian | 2.5851 | 0.5119 | 0.0064 |
| TRPO-SL | 2.5683 | 0.8681 | 0.0071 |
| TRPO-USL | 2.5808 | 0.5921 | 0.0070 |
| TRPO-IPO | 2.5625 | 0.5047 | 0.0071 |
| TRPO-FAC | **2.6599** | 0.4819 | 0.0059 |
| CPO | 2.6440 | 0.2944 | 0.0041 |
| PCPO | 2.6249 | 0.3843 | 0.0052 |
| SCPO | 2.5793 | **0.1427** | **0.0020** |

Table 6: Metrics of three **Point-Pillar** experiments obtained from the final epoch.

(a) Point-Pillar-1

| Algorithm | $\bar{J}_r$ | $\bar{M}_c$ | $\bar{\rho}_c$ |
|---|---|---|---|
| TRPO | 2.6059 | 0.2899 | 0.0026 |
| TRPO-Lagrangian | 2.5772 | 0.1218 | 0.0020 |
| TRPO-SL | 2.5049 | 0.1191 | 0.0014 |
| TRPO-USL | 2.5924 | 0.1483 | 0.0021 |
| TRPO-IPO | 2.6059 | 0.2899 | 0.0026 |
| TRPO-FAC | **2.6362** | 0.0698 | 0.0013 |
| CPO | 2.5464 | 0.2342 | 0.0028 |
| PCPO | 2.5857 | 0.2088 | 0.0025 |
| SCPO | 2.5928 | **0.0040** | **0.0003** |

(b) Point-Pillar-4

| Algorithm | $\bar{J}_r$ | $\bar{M}_c$ | $\bar{\rho}_c$ |
|---|---|---|---|
| TRPO | 2.5958 | 0.4281 | 0.0061 |
| TRPO-Lagrangian | 2.6040 | 0.2786 | 0.0050 |
| TRPO-SL | 2.5417 | 0.2548 | 0.0031 |
| TRPO-USL | 2.5623 | 0.2977 | 0.0063 |
| TRPO-IPO | 2.5958 | 0.4281 | 0.0061 |
| TRPO-FAC | **2.6105** | 0.3223 | 0.0040 |
| CPO | 2.5720 | 0.5523 | 0.0062 |
| PCPO | 2.5709 | 0.3240 | 0.0052 |
| SCPO | 2.5367 | **0.0064** | **0.0005** |

(c) Point-Pillar-8

| Algorithm | $\bar{J}_r$ | $\bar{M}_c$ | $\bar{\rho}_c$ |
|---|---|---|---|
| TRPO | 2.6095 | 3.4805 | 0.0212 |
| TRPO-Lagrangian | 2.6164 | 0.6632 | 0.0129 |
| TRPO-SL | 2.5585 | 1.5260 | 0.0074 |
| TRPO-USL | 2.5836 | 0.6743 | 0.0172 |
| TRPO-IPO | 2.6095 | 3.4805 | 0.0212 |
| TRPO-FAC | 2.5701 | 0.4257 | 0.0068 |
| CPO | **2.6440** | 0.5655 | 0.0166 |
| PCPO | 2.5704 | 6.6251 | 0.0219 |
| SCPO | 2.4162 | **0.2589** | **0.0024** |

Table 7: Metrics of three **Swimmer-Hazard** experiments obtained from the final epoch.

(a) Swimmer-Hazard-1

| Algorithm | $\bar{J}_r$ | $\bar{M}_c$ | $\bar{\rho}_c$ |
|---|---|---|---|
| TRPO | 2.6062 | 0.5326 | 0.0070 |
| TRPO-Lagrangian | 2.6044 | 0.4060 | 0.0056 |
| TRPO-SL | 2.5269 | 10.0374 | 0.0382 |
| TRPO-USL | **2.6296** | 0.3754 | 0.0050 |
| TRPO-IPO | 2.6062 | 0.5326 | 0.0070 |
| TRPO-FAC | 2.5765 | 0.2439 | 0.0041 |
| CPO | 2.6126 | 0.4115 | 0.0049 |
| PCPO | 2.5741 | 0.4670 | 0.0051 |
| SCPO | 2.6006 | **0.0743** | **0.0009** |

(b) Swimmer-Hazard-4

| Algorithm | $\bar{J}_r$ | $\bar{M}_c$ | $\bar{\rho}_c$ |
|---|---|---|---|
| TRPO | 2.5897 | 0.2046 | 0.0033 |
| TRPO-Lagrangian | 2.6128 | 0.3953 | 0.0038 |
| TRPO-SL | 2.5056 | 4.6391 | 0.0206 |
| TRPO-USL | 2.6103 | 0.2260 | 0.0027 |
| TRPO-IPO | 2.5844 | 0.2739 | 0.0033 |
| TRPO-FAC | 2.5984 | 0.1997 | 0.0028 |
| CPO | 2.6023 | 0.1368 | 0.0021 |
| PCPO | 2.5922 | 0.4265 | 0.0033 |
| SCPO | **2.6317** | **0.1082** | **0.0012** |

(c) Swimmer-Hazard-8

| Algorithm | $\bar{J}_r$ | $\bar{M}_c$ | $\bar{\rho}_c$ |
|---|---|---|---|
| TRPO | 2.6322 | 0.4843 | 0.0067 |
| TRPO-Lagrangian | 2.5979 | 0.4205 | 0.0058 |
| TRPO-SL | 2.4930 | 9.6048 | 0.0316 |
| TRPO-USL | 2.6133 | 0.4259 | 0.0059 |
| TRPO-IPO | 2.6322 | 0.4843 | 0.0067 |
| TRPO-FAC | 2.6037 | 0.5606 | 0.0056 |
| CPO | **2.6335** | 0.4201 | 0.0045 |
| PCPO | 2.5895 | 0.7420 | 0.0063 |
| SCPO | 2.5604 | **0.1527** | **0.0030** |

Table 8: Metrics of three **Drone-3DHazard** experiments obtained from the final epoch.

(a) Drone-3DHazard-1

| Algorithm | $\bar{J}_r$ | $\bar{M}_c$ | $\bar{\rho}_c$ |
|---|---|---|---|
| TRPO | 2.3777 | 0.3086 | 0.0014 |
| TRPO-Lagrangian | 2.4149 | 0.0766 | 0.0007 |
| TRPO-SL | 2.4300 | **0.0044** | 0.0004 |
| TRPO-USL | 2.3760 | 0.0690 | 0.0008 |
| TRPO-IPO | 2.3724 | 0.2032 | 0.0011 |
| TRPO-FAC | 2.3856 | 0.0537 | 0.0007 |
| CPO | **2.4464** | 0.0706 | 0.0007 |
| PCPO | 2.1118 | 3.2450 | 0.0015 |
| SCPO | 2.3860 | 0.0423 | **0.0002** |

(b) Drone-3DHazard-4

| Algorithm | $\bar{J}_r$ | $\bar{M}_c$ | $\bar{\rho}_c$ |
|---|---|---|---|
| TRPO | 2.4163 | 0.3008 | 0.0025 |
| TRPO-Lagrangian | 2.4175 | 0.1990 | 0.0022 |
| TRPO-SL | 2.3748 | **0.0529** | 0.0011 |
| TRPO-USL | **2.4658** | 0.1264 | 0.0017 |
| TRPO-IPO | 2.4163 | 0.3008 | 0.0025 |
| TRPO-FAC | 2.3839 | 0.0867 | 0.0015 |
| CPO | 2.3995 | 0.3610 | 0.0026 |
| PCPO | 2.4180 | 1.0088 | 0.0034 |
| SCPO | 2.4034 | 0.0545 | **0.0008** |

(c) Drone-3DHazard-8

| Algorithm | $\bar{J}_r$ | $\bar{M}_c$ | $\bar{\rho}_c$ |
|---|---|---|---|
| TRPO | 2.4206 | 0.4561 | 0.0057 |
| TRPO-Lagrangian | 2.4237 | 0.1962 | 0.0034 |
| TRPO-SL | 2.4255 | 0.1635 | 0.0022 |
| TRPO-USL | 2.4488 | 0.2052 | 0.0037 |
| TRPO-IPO | 2.4206 | 0.4561 | 0.0057 |
| TRPO-FAC | **2.4600** | 0.1069 | 0.0022 |
| CPO | 2.4221 | 0.6941 | 0.0041 |
| PCPO | 2.1837 | 0.5179 | 0.0027 |
| SCPO | 2.3846 | **0.0478** | **0.0012** |

Table 9: Metrics of **Ant-Hazard** and **Walker-Hazard** experiments obtained from the final epoch.

(a) Ant-Hazard-8

| Algorithm | $\bar{J}_r$ | $\bar{M}_c$ | $\bar{\rho}_c$ |
|---|---|---|---|
| TRPO | 2.6203 | 0.1869 | 0.0084 |
| TRPO-Lagrangian | **2.6336** | 0.1667 | 0.0058 |
| TRPO-SL | 2.5522 | 4.1269 | 0.0510 |
| TRPO-USL | 2.6153 | 0.2108 | 0.0083 |
| TRPO-IPO | 2.6197 | 0.1990 | 0.0083 |
| TRPO-FAC | 2.6218 | 0.0955 | 0.0051 |
| CPO | 2.6103 | 0.1330 | 0.0066 |
| PCPO | 2.6281 | 0.1046 | 0.0059 |
| SCPO | 2.5873 | **0.0327** | **0.0021** |

(b) Walker-Hazard-8

| Algorithm | $\bar{J}_r$ | $\bar{M}_c$ | $\bar{\rho}_c$ |
|---|---|---|---|
| TRPO | 2.6471 | 0.3274 | 0.0096 |
| TRPO-Lagrangian | 2.6167 | 0.2194 | 0.0071 |
| TRPO-SL | **2.6476** | 0.9863 | 0.0204 |
| TRPO-USL | 2.6239 | 0.3148 | 0.0095 |
| TRPO-IPO | 2.6397 | 0.3115 | 0.0096 |
| TRPO-FAC | 2.5917 | 0.1283 | 0.0049 |
| CPO | 2.6211 | 0.1779 | 0.0069 |
| PCPO | 2.6410 | 0.2013 | 0.0074 |
| SCPO | 2.5751 | **0.0546** | **0.0029** |

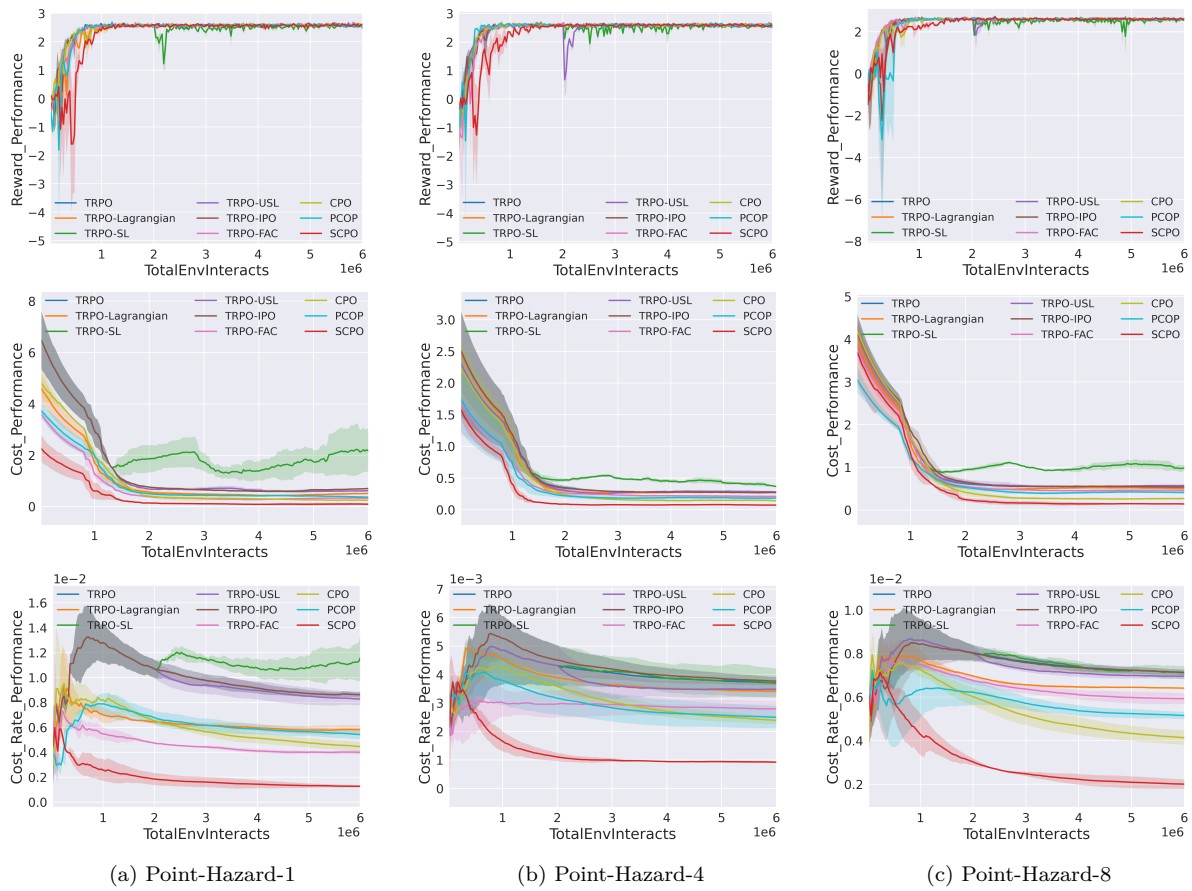

(a) Point-Hazard-1          (b) Point-Hazard-4          (c) Point-Hazard-8

Figure 11: Point-Hazard

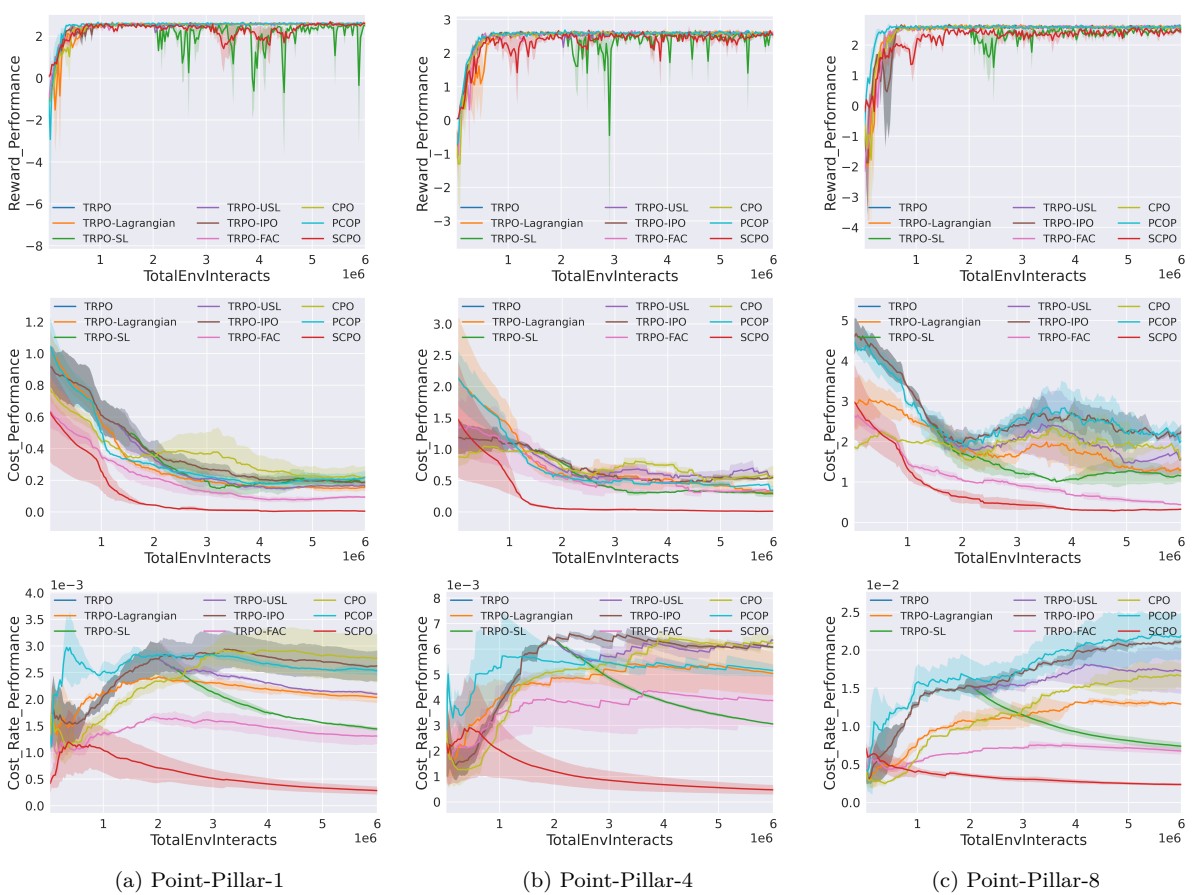

(a) Point-Pillar-1  (b) Point-Pillar-4  (c) Point-Pillar-8

Figure 12: Point-Pillar

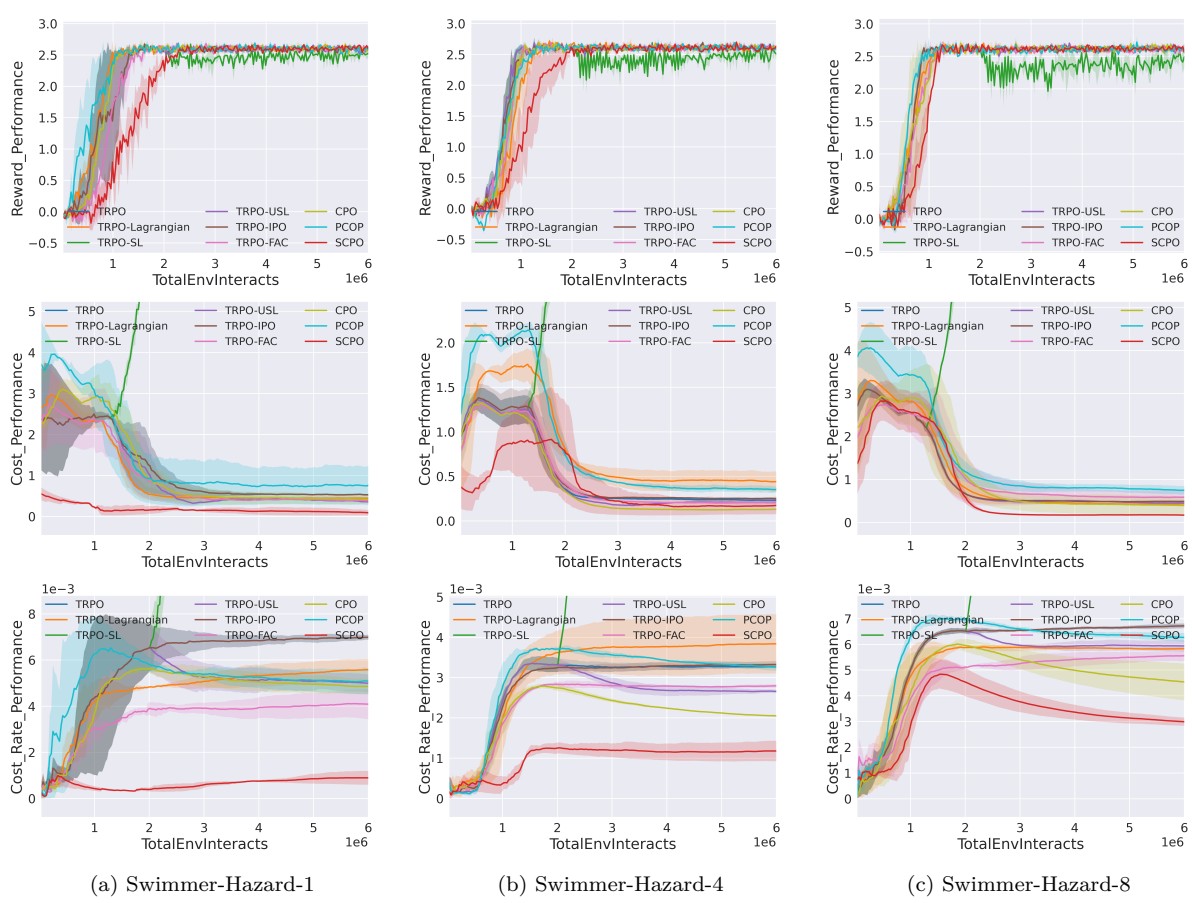

(a) Swimmer-Hazard-1     (b) Swimmer-Hazard-4     (c) Swimmer-Hazard-8

Figure 13: Swimmer-Hazard

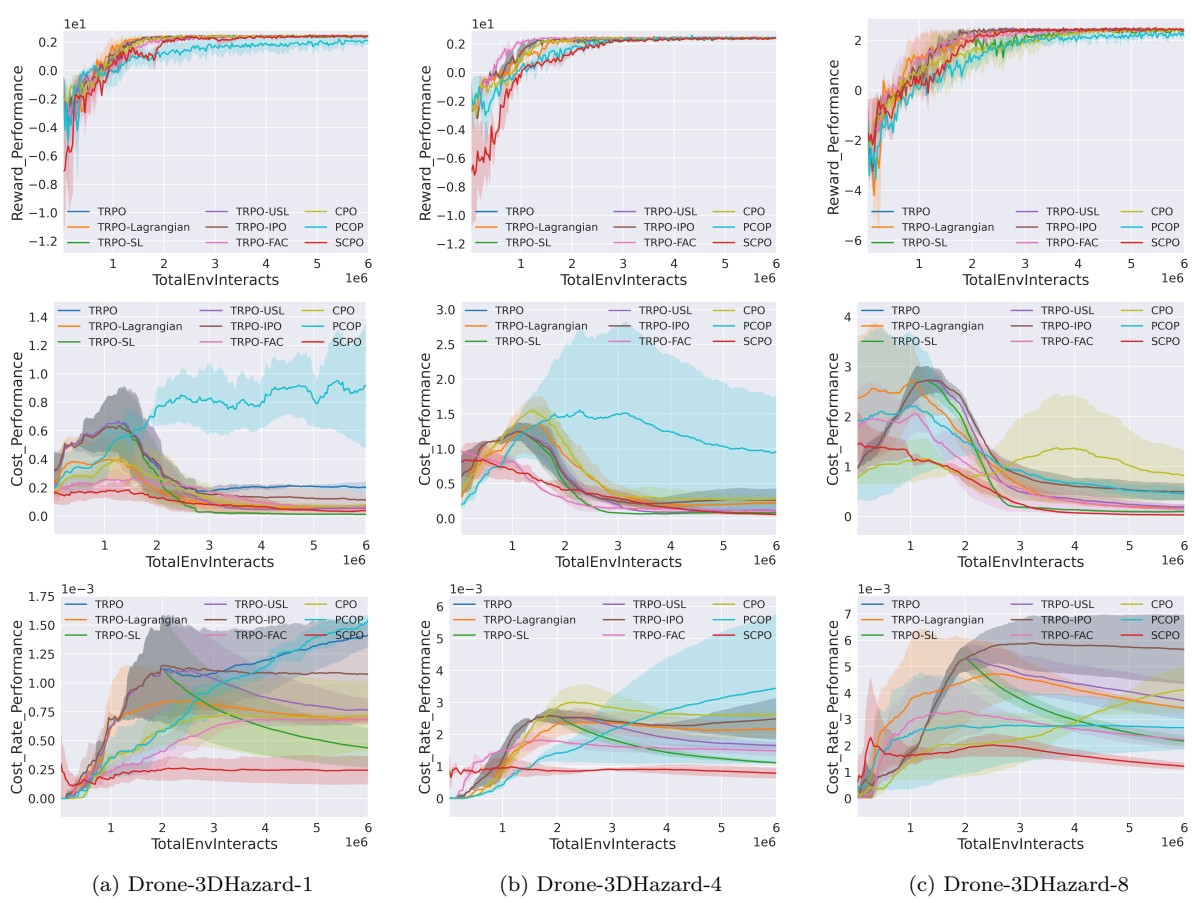

(a) Drone-3DHazard-1    (b) Drone-3DHazard-4    (c) Drone-3DHazard-8

Figure 14: Drone-3DHazard

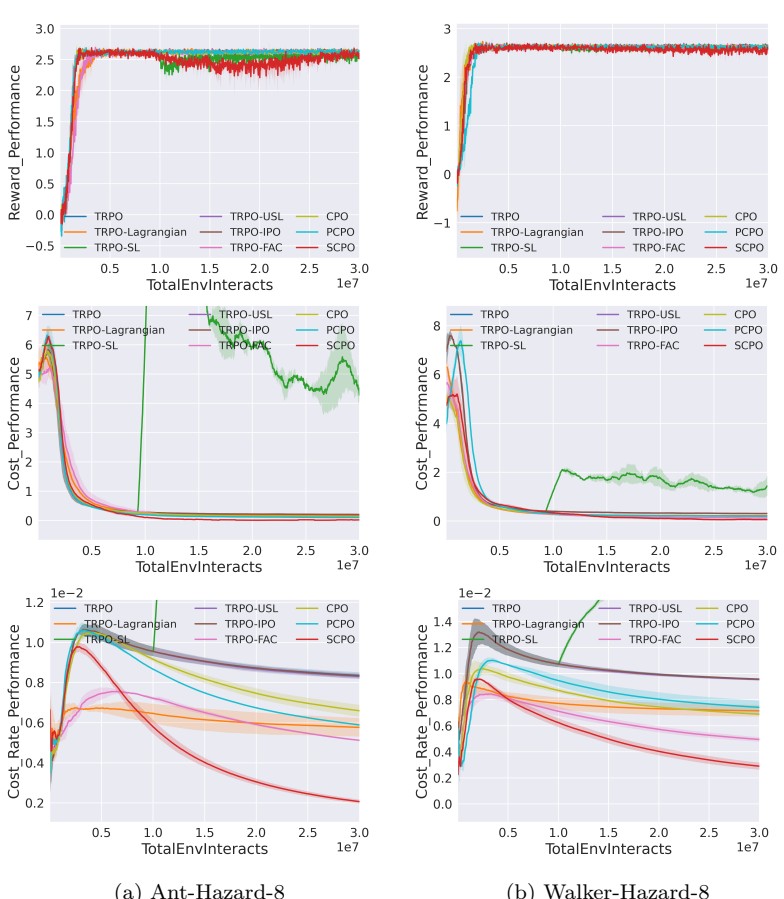

(a) Ant-Hazard-8    (b) Walker-Hazard-8

Figure 15: High dimensional hazard tasks

