# OpenReview forum: "State-wise Constrained Policy Optimization"
_TMLR — Accepted by TMLR_

### Review · Reviewer_u6rH · 2023-12-17

**Summary Of Contributions:**

This paper tackles reinforcement learning in Markov decision processes with (soft) constraints that must be satisfied at each point in time.  This is as opposed to existing ``constrained'' methods, where the (soft) constraints must only be met in expectation across time.  A new formalism, bound, and learning method are introduced for operating in this regime.  The constraints are re-formulated so that they can be efficiently estimated.  Some results on a safety-oriented benchmark are shown, and the algorithm appears to perform as desired on these tasks.

**Audience:**

Yes

**Claims And Evidence:**

Yes

**Requested Changes:**

1.  Fix all Minor Weaknesses.
2.  If you could evaluate/probe the tightness of the bounds then that would be super helpful.
3.  More evaluation/reflection on whether imposing constraints per-timestep is (a) well-founded mathematically (c.f. 2),  (b) actually corresponds to something better than alternatives (e.g. including just a large negative reward penalty for infractions),  (c)

**Strengths And Weaknesses:**

## Strengths
- The paper is very well written.
- The problem formulation _seems_ novel, and, more importantly, could be a useful adjustment to existing formulations.
- The method seems to perform well, and has nice tie-ins to existing results (e.g. TRPO).
- The paper as a whole is about the right level and tone for TMLR.

## Weaknesses

1.  I don't know if the formulation in (6) does really does correspond to the intended objective.  (6) states that the policy only needs to satisfy the constraint in expectation at each timestep.  This is a small and slightly disappointing departure from the setup, eg the final paragraph of the introduction.
    - For tasks that have different intermediate "horizons" as well, this per-timestep interpretation gets a little warped, e.g. if one environment is completed in H/2, whereas another is completed in H, then comparing rewards/costs-to-go on a per-timestep basis is a little tricky.

2.  Beyond this, the technical contribution is minor, in my opinion.  The MMDP (and even CMDPs) is just an MDP under some constraints on the policy set.  I don't really see that as a ``contribution''.  The bound in (12) is similar in its construction to the TRPO bound.  I expect this bound to be _very_ loose (this is not checked in the empirical evaluation).  The reformulation to allow function approximations to be used tractable is nice, but the accuracy or sensitivity to these function approximations isn't really established or investigated.

3. I'm also not sure if the empirical evaluation really zeroes in on the actual pre-existing deficiency.  They sort of just seem to show that the proposed algorithm ``works''.  Extra evaluation of small-scale failcases that can be really well understood and analyzed would have helped convince me that the proposed formulation is unequivocally addressing the deficiency.  Maybe an extra experiment where there is a hazard that appears at a specific timestep and hence amortizing infractions over the trajectory leads to more infractions?

## Minor Weaknesses.
- Only proper nouns should be capitalized ("Markov Decision Process" -> "Markov decision process").
- Use \citet and \citep where appropriate.
- It is not clear in a lot of the expressions what the subscripts are indexing.  More consistent inclusion of $i \in 1,\ldots,m$ would be helpful.
- Figures and tables (e.g. Fig 3) should be floated to the top of pages.
- "Cost Rate" in Figure 4 is undefined.
- The transition from (2) to (3) is unclear -- please be more precise.
- (maybe personal preference) I don't especially like for formatting/content of Remarks 1 and 2.  If these are important points, then maybe they should be included in the proposition or theorem somehow?  If these are direct results of the proposition, then they probably should be just normal text.  Remarks are often reserved for non-obvious follow-ons.  RIght now they disrupt the flow.

## Summary.
I think this paper is at the requisite level for TMLR.  I don't think the contribution is huge, but i think is relatively self-contained and is a useful method/exposition to have in the public domain.  The results are sufficient, even if they don't bowl me over.  I have a number of small requests that need to be fixed prior to inclusion.  I then have a few small suggestions, but these are not hard conditions for inclusion.

Good work, and good luck.

---

> ### Author Response · Authors · 2024-01-13
> **Reply to Requested Changes**
>
> We extend our sincere gratitude for your thorough comments on our paper. We have updated the paper draft according to your comments, the new contents are highlighted in red color. Below, you will find our responses to your inquiries.
>
> **Requested Changes 1**: Fix all Minor Weaknesses.
>
> **Answer 1**: Thanks! We have addressed all minor weaknesses in the current version. $$ $$
>
> **Requested Changes 2**: Evaluate the tightness of the bounds.
>
> **Answer 2**: Thanks! The analysis of the tightness of the bounds is now presented in Section 6.2. $$ $$
>
> **Requested Changes 3**: More evaluation on imposing constraints per-timestep both mathematically and empirically.
>
> **Answer 3**: We have updated the manuscript to include (i) The mathematical superiority of the constraints on state-wise cost at the end of Section 3.2. (ii) The ablation study on methods with different negative reward penalties in Appendix F.4.

---

### Review · Reviewer_VjA5 · 2023-12-18

**Summary Of Contributions:**

The paper proposes a new RL algorithm for constrained policy optimization, which aims to minimize constraint violations while preserving high performance.

The method relies on modifying the standard Markov Decision Process formulation to include a term for the maximum cost encountered so far in a trajectory - or more precisely, for statewise increments in this maximum (if any). The aim is to maximize expected discounted returns, while keeping maximum costs encountered under a minimum.

The algorithm also includes a trust-region penalty. Theoretical guarantees on the constraint violations and performance are derived. Experiments show that the method can reduce constraint violations in comparison to various alternative methods, while preserving high performance.

More detailed summary:

The introduction specifies the goal: "design a policy improvement step that not only guarantees worst-case performance degradation but also ensures state-wise cost constraints."

The language is that the method provides hard guarantees on the maximum cost at each individual step, but the equations are all about expectations. This poses a difficulty in interpreting the paper. See below.

Eq. 4: d_i is a maximum acceptable [either max or sum-of-discounted] cost (as estimated by function C_i) over an episode.

M seems to represent "the maximum transition cost encountered so far in a given trajectory" (That's how I interpret Eq. 8 and the definition of M_it just below it). Di represent "how much larger the cost at this particular transition is than the previous maximum", thresholded at 0.

Eq. 9 suggests that the goal is to keep the expected sum of D below a certain maximum.

Eq. 11: you want the new policy to satisfy the constraint *and* to be within a small Dist (to be defined) of previous policy.

Eq. 12 is the actual equation for the proposed method (IIUC), including a trust-region penalty.

Prop 1 seems to say that a policy found by satisfying Eq. 12 will meet the maximum-cost-increment criterion (again, in expectation!) (Presumably this assumes that the previous policy pi_k also met the criterion).

End of section 5 discusses some approximations that make the method practical.

Figure 4 seems to show that SCPO minimizes the costs better, while achieving competitive returns.  Figure 5 shows actual "state-wise costs".

**Audience:**

Yes

**Broader Impact Concerns:**

I do not see any broader impact concerns.

**Claims And Evidence:**

Yes

**Requested Changes:**

There should be a clarification in the main text of what exactly the objective is. The text of the article suggests that the method seeks to provide constraints on the maximum cost encountered so far in any given trajectory - in the authors' own words, ""to persistently satisfy a hard cost constraint *at every step* (as opposed to cumulative costs over trajectories)". However, the equations are all about  expected average over many trajectories, which is precisely equal to these "cumulative costs" up to a divisive constant. This apparent discrepancy should be resolved in the main text.

Other requested changes:

p.1 degredation -> degradation

Eq. 2 and 3 define the same quantity J_Ci(pi) as two different things!

Second paragraph of p.5 has the Schulman citation with parenthesis in the wrong place (pro-tip: define \cite as \citep in your latex file). (Nitpick: TRPO isn't very "recent"!)

"is bounded" -> "to be bounded". "solving following" -> "solving the following".

In Eq. 12, in the last line, why does it include a term for the expected advantage? The equation seems to say that candidate policies with high advantages must have a lower expected cost J_D to be accepted - why is that?

P. 7: Citation to Ray et al. again has wrong parenthesis. There should be parentheses around the "Figure XX" references.

Figure 4 seems to show that SCPO minimizes the costs better, while achieving competitive returns. Though again, these are some averages.  Figure 5 shows "state-wise cost", which would better match what the text of the article proposes (in the authors' words, "hard cost constraint at every step (as opposed to cumulative costs over trajectories"). How exactly is it computed? That is, for each point in Fig.5, what is the sample over which we take the maximum?

**Strengths And Weaknesses:**

Strength: the problem is important, the method seems novel and interesting (as far as I could understand it) and the results look promising.

Weaknesses: The main difficulty is that there seems to be a discrepancy between what is promised in the text, and what the equations actually seem to do. Briefly, the text suggests that the method can provide hard guarantee on the maximum state-size cost ever encountered, but in practice the equations are all about expectations (i.e. averages) over multiple trajectories.

Top of p. 4, we are told that the goal is "to persistently satisfy a hard cost constraint *at every step* (as opposed to cumulative costs over trajectories)"... (emphasis original!).

....but then Eq. 6 (and following) denote the *expected* cost over all transitions from all trajectories from a given policy! This should be equal to the "cumulative costs" (not just within an episode, but over all encountered episodes) divided by sample size.

Similarly,  Eq. 9 is an expectation (of maximum cost) over all trajectories! A policy could keep this quantity under a maximum w_i (as in Eq. 10), while still encountering arbitrarily high costs (much larger than w_i) in rare transitions/trajectories. Thus it is not clear exactly what guarantee is provided - about absolute maximum, or about expectations over many trajectories.

---

> ### Author Response · Authors · 2024-01-13
> **Reply to Requested Changes**
>
> We extend our heartfelt thanks for your valuable feedback on our paper. Your time and effort invested in reviewing our work are sincerely appreciated. We have updated the paper draft according to your comments, the new contents are highlighted in red color. Below, you will find our responses to your questions.
>
>
>
> **Weakness and Requested Changes 1**: Clarification of the objective of this paper and what guarantee is provided.
>
> **Answer 1**: Thanks for your suggestion! We have updated our wording with ``satisfaction of state-wise hard constraint`` into ``state-wise constraint satisfaction in expectation`` in Section 1 (Introduction) and Section 3.2. This update offers a more precise description of our objective and theoretical guarantee. Additionally, at the end of Section 3.2, we provide additional analysis elucidating why SCMDP produces safer policy than satisfying cumulative constraint in CMDP, where cumulative constraint means to restrict the discounted summation over cost along trajectory. $$ $$
>
>
> **Other Requested Changes**: Fix all minor flaws.
>
> **Answer 2**: Thank you! We have addressed all minor flaws you pointed out in the current version. More explanation of the metric in Fig.5 (Fig.6 in the new version) is presented in Section 6.2.

---

### Review · Reviewer_qZLB · 2024-01-03

**Summary Of Contributions:**

The authors propose SCPO, a policy optimization algorithm for solving RL tasks with state constraints. They propose a new formulation that constrains the expected maximum cost incurred over the trajectory rather than expected cumulative cost. Theoretical analysis is provided for this formulation. Their empirical algorithm solves a convex approximation of the objective, and use some implementation tricks like under-sampling to improve  cost value function learning. SCPO shows gains over baselines in robotic tasks in the benchmark SafetyGym.

**Audience:**

Yes

**Broader Impact Concerns:**

None.

**Claims And Evidence:**

Yes

**Requested Changes:**

## Requested Changes
See above for details.
- More through analysis / ablation of the under-sampling trick wrt baselines and SCPO itself. I ask this because I don't want to have a method whose claimed algorithmic gains come primarily through an implementation trick.
- It would be nice to show (positive or negative) results on harder tasks. The current tasks are just barely over the bar (marginal but consistent gains) for demonstrating SCPO's effectiveness. Even negative results for SCPO or methods in general will be useful, since I think the SafetyGym tasks seem too easy. This would outline a good future direction for future work; I do not want future work to waste their efforts chasing marginal gains on a saturated benchmark.

## Clarification Questions
**Why does maximum cost formulation result in empirically lower average cost over expected cost formulation baselines?** SCPO seem to have lower average cost (cost rate) over training. Since SCPO constrains the expected maximum cost over the trajectory, it could pick policies that have relatively high average cost per timestep but low maximum cost over the entire trajectory.  For example, it could have a trajectory with costs (1, 1, 1, 1 .... 1.01). The maximum cost of this trajectory is 1.01, but all other steps still have high costs. Would this trajectory be considered equal to a trajectory with all zeros but one step with 1.01 cost?

Can this work with costs with negative values? The method seems to assume non-negative cost increments.

Eq 2 / 3 seem to use the same left hand variable, but have different right hand definitions? Are they equivalent?

Sec 3.3 is rather laconic in setting up its definitions in the first paragraph. Could the authors attempt to add a bit more explanation in the MMDP explanation, considering the importance of this setup?

Following that intuition, we define a novel Maximum Markov-Decision Process (MMDP), ...
The cost increment functions definition should it be: $D_i : (S, M^i) \ldots$?

Augmented state definition seems a bit imprecise. Is it only part of $\mathcal{M}^m$ or any arbitrary $\mathcal{M}$?

**Strengths And Weaknesses:**

[S1] The SCPO formulation is well-motivated and has theoretical analysis (although I did not extensively check the proofs).

[S2] Good performance on SafetyGym benchmark
- Consistent but often marginal gains, in rewards and episodic costs, over all tasks.
- Definitively beats methods in average cost over training.

[S3] Contributes harder tasks to the SafetyGym benchmark, although they still seem too easy (see below).

[W1] **Importance of under-sampling trick  for value function learning**: This value function training trick can be applied to the other baselines right? It would be good to see some more ablations into this trick. The authors show an ablation for SCPO without undersampling for one environment. It would be good to show this ablation for all other environments.

This could be an important empirical contribution for safe RL actor-critic methods if it turns out to be helpful in general. It would also more clearly show the contribution of the SCPO update rule

[W2] **Benchmark seems too easy:** Benchmark seems rather easy. The authors claim they solve high-dimensional tasks giving the impression of difficulty, but I think this is inaccurate. For the 2 hardest tasks, nearly all methods achieve very similar reward and episodic cost metric values. This doesn't seem very hard to me.

It would be interesting to see some harder higher dimensional tasks like humanoid. The benchmark is also overfit to simple mujoco locomotion tasks, which is well-known to be "easy" in terms of exploration and dynamics. Adding more complex tasks, like dexterous manipulation, quadrupedal / humanoid locomotion, would be very interesting.

[W3] **No discussion on resource usage and computational complexity.** Please report GPU / CPU resources needed for the method, baselines, time for experiments, etc. Next, what algorithmic and runtime differences does SCPO have with baselines?

---

> ### Author Response · Authors · 2024-01-13
> **Reply to Requested Changes and Clarification Questions**
>
> Thank you for your valuable feedback on our paper. We sincerely appreciate your time and effort in reviewing our work. We have updated the paper draft according to your comments, the new contents are highlighted in red color. Please find our answers to your questions below.
>
> **Weakness 1 and Requested Change 1**: Why is the under-sampling trick much more important for SCPO?
>
> **Answer 1**: We have added a thorough analysis and ablation of under-sampling trick wrt. baselines and SCPO within “Ablation on Sub-sampling Imbalanced Cost Increment Value Targets” in section 6.2. $$ $$
>
>
> **Weakness 2 and Requested Changes2**: Benchmarks seem too easy.
>
> **Answer 2**: Thanks! We have included additional results on dexterous robot arm manipulation and safe humanoid locomotion. Specifically, we updated section 6.1, figure 5, and *High Dimensional System* in section 6.2.  $$ $$
>
>
> **Clarification Questions1**: More explanation of MMDP.
>
> **Answer 3**: Yes, these two trajectories are equal in MMDP since their maximum cost is the same. SCPO has a low cost rate (state-wise cost) $\rho_c$ since SCPO directly restricts state-wise cost while other CMDP baselines have no control over it. SCPO has a low average episodic sum of cost ($M_c$) since it is proportional to $\rho_c$ and average episodic length, where average episodic length is roughly the same across different baselines. $$ $$
>
> **Clarification Questions2**: Can SCPO work with costs with negative values?
>
> **Answer 4**: Yes! SCPO works with costs with negative values. It's important to note that the cost increment is computed as the relative increment between consecutive costs, which can be either positive or negative. $$ $$
>
> **Clarification Questions3**: Eq 2 / 3 confusion.
>
> **Answer 5**: We have fixed this issue. Now only Eq 2 is included.  $$ $$
>
> **Clarification Questions4**: More explanation of MMDP intuition.
>
> **Answer 6**: Thanks for this suggestion! We have added a paragraph of explanation in Section 3.3, and included a visualization to help understanding in Figure 1.  $$ $$
>
> **Clarification Questions5**: Cost increment function input.
>
> **Answer 7**: We have updated the definition of $D_i$ input space to make it more precise in Section 3.3.  $$ $$
>
> **Clarification Questions5**: Imprecise augmented state definition.
>
> **Answer 7**: We have updated the definition of $\mathcal{M}_m$ to make it more precise in Section 3.3.

---

> > ### Comment · Reviewer_qZLB · 2024-01-15
> > **More details on Arm and Humanoid experiments needed**
> >
> > I appreciate the Arm and Humanoid task additions.
> > Could you provide some more details? For example, the state space, the rewards, etc. It is unclear what the task of Arm and Humanoid are. Is the Arm task just to move the robot gripper to some position in 3D? Then that is not dexterous manipulation, that is a trivial manipulation task. Dexterous manipulation is usually characterized by a high DoF robot hand that is moving some object.  It has complicated dynamics and is known to be a hard RL task.  See "Manipulate Block" in [gymnasium robotics](https://robotics.farama.org/envs/shadow_dexterous_hand/index.html) for a dexterous manipulation task.
> >
> > Next, can you provide episode visualizations of the methods for each task? I am somewhat surprised all methods seem to work on the Humanoid task. Do all policies learn some locomotion behavior?

---

> > > ### Author Response · Authors · 2024-01-16
> > > **Reply of details on Arm and Humanoid experiments**
> > >
> > > **Q1**: Details about Arm and Humanoid Task
> > >
> > > **A1**: Both Arm and Humanoid Tasks are performing safe goal reaching. We have summarized the detailed reward function design, state space (internal state space and external state space) in Appendix F.1 (Environment Settings). $$ $$
> > >
> > > **Q2**: Dexterous manipulation of Arm Task
> > >
> > > **A2**: Thank you for bringing this to our attention! The Arm Task in our study involves safely moving the robot end effector towards a 3D position. In the revised draft, we have adjusted the wording to replace all instances of ``dexterous manipulation`` with "robot arm goal reaching" to avoid any potential confusion or overstatement. Our current implementation, based on SafetyGym, focuses on obstacle avoidance in the context of robot locomotion. While safe robot hand manipulation is an exciting avenue, its inclusion would necessitate significant modifications to the SafetyGym engine, which is beyond the scope of this paper. We express our eagerness to explore safe robot hand manipulation in future work. Your understanding and guidance are highly appreciated.  $$ $$
> > >
> > > **Q3**: Visualization of each task
> > >
> > > **A3**: We have created videos showcasing trained SCPO policy for each task. These videos can be found in the `video`  folder within the updated supplementary materials.  $$ $$
> > >
> > > **Q4**: All methods work in Humanoid Task?
> > >
> > > **A4**: Yes, all the methods can succeed with locomotion for goal reaching, with variations in their safety performance. This is because we ease the learning process for the humanoid by excluding the initial phase of learning to stand up. This phase typically requires a considerable amount of training time and is not directly relevant to safe behavior. To achieve this, we constrain the humanoid's upper body to remain above a certain height threshold from the ground. As a result, each algorithm can concentrate on developing its own safe locomotion policies.

---

### Decision · Action_Editor_QGXy · 2024-03-04

**Recommendation:** Accept as is

**Comment:**

As detailed above, the paper now makes a clear and consistent case for its proposal of a new algorithm suited to a specific notion of safety in reinforcement learning. It is not without limitations as pointed out above in the "claims and evidence", but it is the AE's opinion that the findings remain valuable to share to the TMLR audience.

**Audience:**

Yes, the TMLR audience includes substantial communities of researchers in reinforcement learning, safe learning, and robot learning, many of whom will be interested in knowing the findings of this paper.

**Claims And Evidence:**

The paper proposes a modified reinforcement learning objective and optimization procedure to deal with state-wise constraints, evaluated on standard safety-gym tasks as well as some more challenging new tasks in the rebuttal phase. Another improvement in the rebuttal phase is the refinement of the claims to be more precise about the specific nature of the state-wise constraint satisfaction claim, which is now more limited after reviewers pointed out the discrepancy between claims and the actual method and results. Overall, the paper during discussion and rebuttal phase has improved so that it now indeed presents clear and accurate evidence of its claims.

On the negatives side, the size of the gains is not always consistent or large, and there isn't a clear study of what causes gains to be larger in some cases than others. It is also not very well-explained why the notion of safety that this paper tackles is a natural one that practitioners for some realistic problems should care about.

Overall however, it is clear that the contribution here is valuable to share to the TMLR community.